# Noncollinear harmonic spectroscopy reveals crossover of strong-field effects

Jicai Zhang [1] ✉, Xiulan Liu[2,3], Tien-Dat Tran [1], Wenqi Xu[1], Wenhao Yu[1], Chong Zhang[1], Ziwen Wang[1], Lei Geng [2], Jianing Zhang [2], Liang-You Peng [2,3,4] ✉, Stanislav Yu. Kruchinin [5] ✉ & Tran Trung Luu [1] ✉

The ability to control electron motion with light fields represents a transformative frontier in modern physics, enabling dynamic manipulation of material properties at ultrafast timescales. Yet, the complex interplay between light and excited carriers—via mechanisms such as the AC Stark effect, field-induced coupling of excitonic and Bloch states, the dynamical Franz-Keldysh effect, and the ponderomotive effect—continues to challenge our understanding of quantum systems driven far from equilibrium. Here, we establish non-collinear harmonic spectroscopy as a powerful technique for initiating, tracking, and steering femtosecond carrier dynamics across the energy landscape in the dielectric $SiO_2$ crystal. Combining rigorous numerical simulations with analytical theory, we identify the main mechanisms responsible for the crossover of different strong-field phenomena, which leads to the delay-dependent energy shift of excitonic and Bloch states. This control over the electronic and excitonic states opens new opportunities for tailoring carrier dynamics in quantum materials, paving the way for next-generation optoelectronic and nanophotonic technologies.

Strong light-matter interaction creates non-equilibrium states of matter, presenting a remarkable avenue to explore microscopic phenomena and macroscopic properties beyond the reach of conventional materials[1,2]. Controlling electron dynamics in solids on femtosecond (1 fs = $10^{-15}$ s) to attosecond (1 as = $10^{-18}$ s) timescales holds great potential for advancing ultrafast optoelectronics[3] and exploring non-equilibrium many-body physics with unprecedented accuracy. For instance, it opens exciting possibilities for dynamically tailoring the electronic structure using Floquet engineering[4–7], generating higher-order harmonics[8–19], inducing phase transitions[20,21], exploring superconductivity[22], topological phenomena[6], non-Markovian dephasing[23,24], and anomalous Hall effect[25] in quantum materials on ultrafast timescales.

On the other hand, miniaturizing components has become a major challenge in the microelectronics industry. Currently, 2.5-dimensional and 3-dimensional packaging offer new pathways toward higher performance and energy efficiency, yet they require a high density of interconnections between logical and memory units[26]. This high density is achieved using interposers, the thin layers made of silicon and drilled with the deep reactive ion etching method[27]. The industry is now exploring the use of laser-drilled thin glass interposers, which are more affordable and easier to process[28]. These developments and novel applications of silicon dioxide have attracted significant attention from both academic and applied researchers studying interactions of materials with strong electric fields.

By carefully tuning the parameters of the optical fields, it is possible to selectively manipulate the relative contributions of interband and intraband processes, allowing for dynamic control over the electron energy landscape. Ultrafast control of electron energies in quartz has several promising applications across fields that rely on precise

[1]Department of Physics, The University of Hong Kong, Hong Kong SAR, China. [2]State Key Laboratory for Mesoscopic Physics and Frontiers Science Center for Nano-optoelectronics, School of Physics, Peking University, Beijing, China. [3]Beijing Academy of Quantum Information Sciences, Beijing, China. [4]Collaborative Innovation Center of Extreme Optics, Shanxi University, Taiyuan, China. [5]Microsoft Austria, Vienna, Austria. ✉e-mail: jczhang@hku.hk; liangyou.peng@pku.edu.cn; stanislav.kruchinin@microsoft.com; ttluu@hku.hk

electronic manipulation and switching[29–32]. These advancements present new routes for understanding and harnessing the rich and dynamic behavior of quantum materials, paving the way for the development of novel functionalities and applications[33,34]. On the other hand, in the presence of a strong electric field, significant amounts of charge carriers could be excited into the high-energy states, leading to the manifestation of various many-body effects, such as Auger scattering[35,36], avalanche breakdown[29,37], carrier-carrier[38–40] and carrier-phonon scattering[41–43], etc. Notably, under intense photoexcitation, electron–phonon coupling can dominate the relaxation

pathways of carriers, leading to enhanced optical conductivity[44]. The intricate interplay among particle dynamics, excitonic effects, and phonon interactions produces a highly nonlinear and complex evolution of the system. Despite these advancements, achieving precise control over electronic dynamics and decomposing these constituent effects remains a significant challenge.

In the present work, we utilize the coherent response of noncollinear high-harmonic spectroscopy to delve into the femtosecond and pathway-resolved control of carrier energies. With experimental data, numerical, and analytical derivations, we study comprehensively the

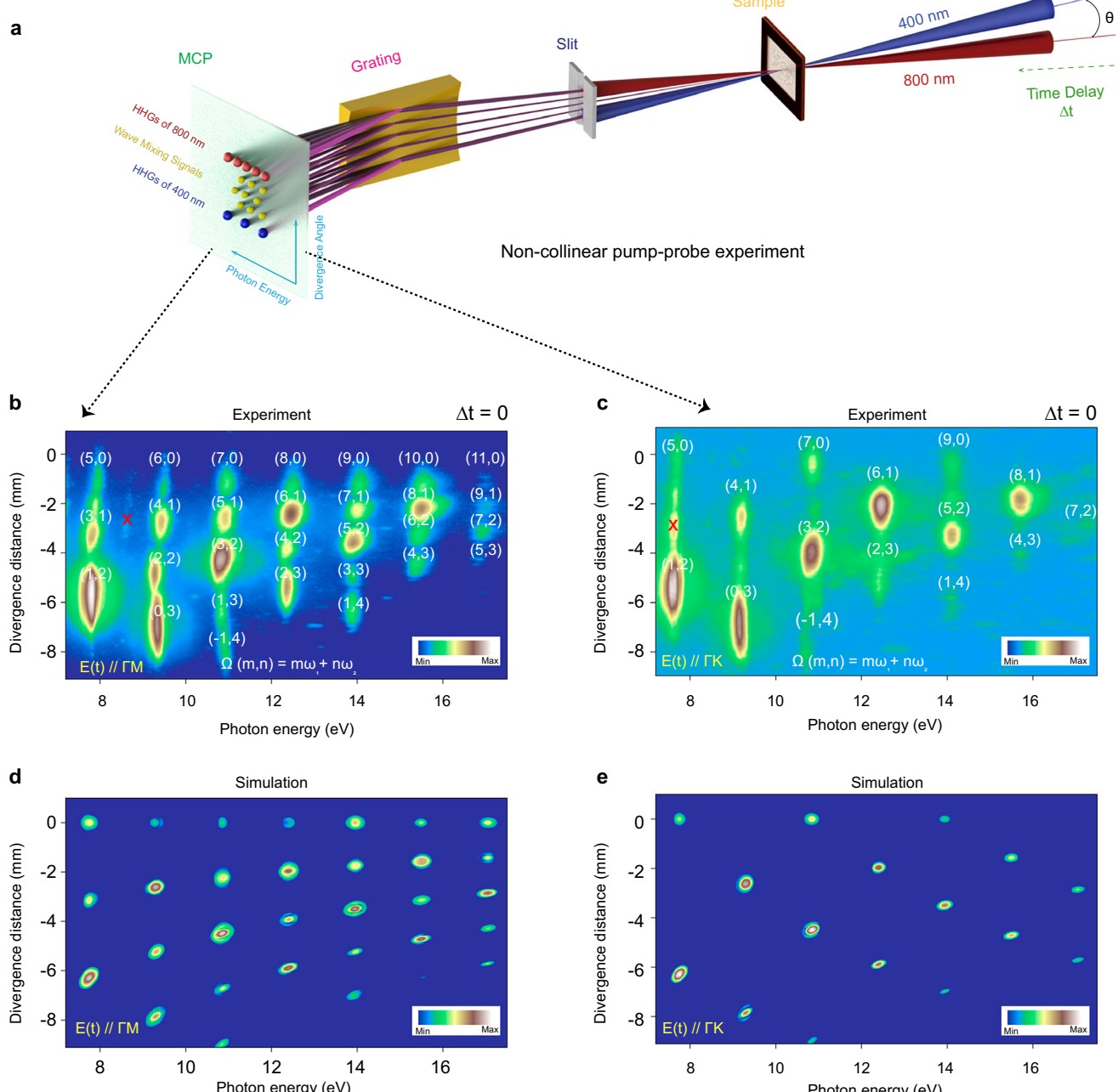

**Fig. 1 | Spatiotemporally resolved ultrahigh-order wave mixing spectra.**
**a** Schematic representation: The $\omega_1$ beam is positioned in the horizontal plane, while $\omega_2$ the beam points downwards at an angle of $\theta$ - 18 mrad, forming a plane parallel to the entrance slit of the XUV spectrometer. **b**, **c** Ultra-high-order wavemixing spectra generated from the superposition of an intense pump fundamental $\omega_1$ (800 nm) at zero time delay and the second harmonic weaker probe field $\omega_2$ (400 nm) in the α-quartz sample, with both pulses linearly polarized to the Γ-M (**b**)

and Γ-K (**c**) directions of the Brillouin zone, respectively. The number inside the bracket $(m, n)$ represents the number of absorbed photons from the pump and probe pulses, respectively, i.e., $\Omega(m, n) \propto (m\omega_1 + n\omega_2)$. **d**, **e** represent the noncollinear simulations considering inversion symmetry parallel to the two electric fields, either in the Γ-M (**b**) or Γ-K (**c**) directions of the Brillouin zone. The color bars are normalized to their maximum harmonic yields and presented on a logarithmic scale. The symbol × indicates the second-order diffraction signal.

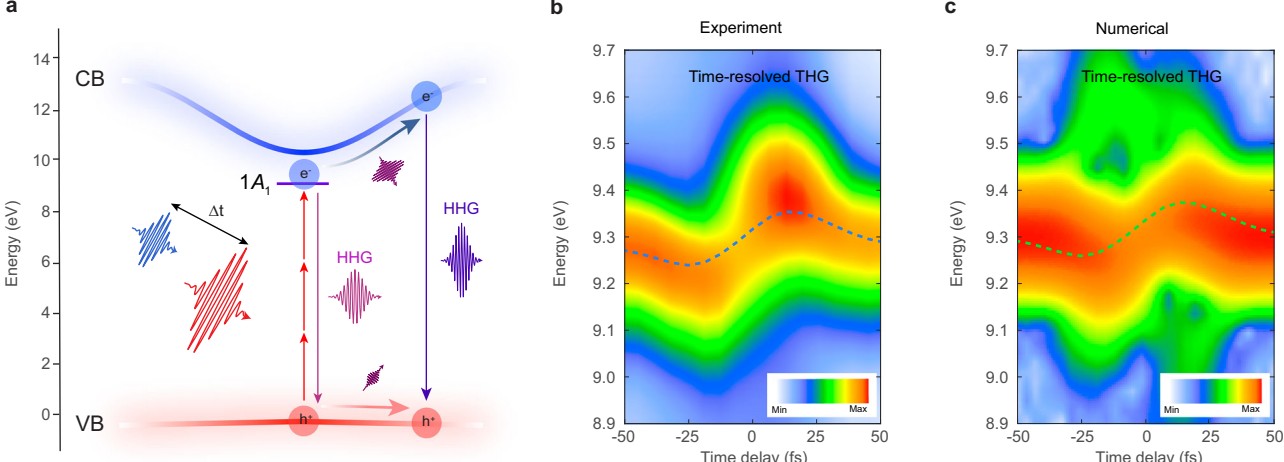

**Fig. 2 | AC Stark shift and dynamical Franz-Keldysh effect revealed by time-resolved non-collinear spectroscopy. a** The main mechanisms of HHG in solids involving interband polarization and intraband currents within a pump-probe geometry. Note that $\Delta t > 0$ indicates that the pump pulse precedes the probe pulse. The pulse duration of the pump and probe is 25 and 28 fs, respectively. **b, c** The experimentally measured and numerically simulated time-resolved (0, 3) (third harmonic, H3) spectra of the probe pulse. The coloured dashed line in the measured and simulated spectra depicts the spectral shift of the third-harmonic energy $\Delta E^{(0,3)}$. The laser pulses are polarized along the $\Gamma$-K direction of the Brillouin zone for both the measured and simulated spectra. The color bars are normalized to their maximum harmonic yields and presented on a logarithmic scale.

dynamic energy shifts of Bloch states and excitonic levels. For negative time delays, when the majority of carriers remain in the valence band, the third-harmonic signal of the probe pulse features the redshift due to the pump-field induced renormalization of the electronic sub-system: the dynamical Franz-Keldysh effect and the excitonic Stark effect. For positive time delays, when a significant number of carriers is excited to the excitonic and Bloch states, we observe the blueshifts due to virtual transitions from the excitonic to Bloch states and the ponderomotive effect. These results demonstrate opportunities for the realization of spectrum-based femtosecond optical switches and for exploring the non-equilibrium many-body phenomena in solids. Our research deepens the understanding of light-matter interactions and control of electronic states with unprecedented precision.

## Results

Figure 1a illustrates the generation of ultrahigh order wave-mixing photons resulting from the superposition of an intense fundamental field with a cycle-averaged intensity of $I_1 \approx 1 \times 10^{13}$ W/cm² and a weak second-harmonic field with an intensity of $I_2 \approx 1 \times 10^{12}$ W/cm². This phenomenon occurs when the two fields are arranged at the crossing angle of $\theta \approx 18$ mrad and time zero, for details see Supplementary Information (SI), Section 1.1. A similar experimental arrangement has been previously discussed in the context of ultrahigh wave-mixing measurements in the gas phase[45]. Compared to conventional collinear pump-probe geometry, this noncollinear configuration offers several advantages, including background-free measurements by avoiding two-photon sideband absorption and energy overlap under two-color field conditions, as well as the capability for momentum- and parity-resolved high-harmonic generation (HHG) spectral measurements.

Notably, owing to the slight inclination of the two pulses, as shown by the solid lines in Fig. 1a, not only their respective harmonics but also their combined wave-mixing frequencies are recorded. The $\alpha$-quartz crystal, composed of silicon (Si) and oxygen (O) atoms, belongs to the trigonal crystal system, with space group $P3_121$ for left-handed and $P3_221$ for right-handed quartz, and lacks spatial inversion symmetry, leading to the emergence of significant Berry curvature responsible for the generation of even harmonics with polarization perpendicular to the driving field[12]. In the case of our noncollinear geometry, the

inversion symmetry of the electric field cannot be perfectly broken. Yet the $\alpha$-quartz crystal already has a broken inversion symmetry along the $\Gamma$-M direction of the Brillouin zone. As a result, as shown in Fig. 1b, we observe the emission of an even total number of photons, represented as $m + n =$ even, such as ($m = 2$, $n = 2$), (4, 2), (6, 2), and (8, 2) mixing photons, deviating from the parity conservation law. To provide a similar observation, we recorded the ultrahigh wave-mixing spectrum in the $\Gamma$-K direction, as depicted in Fig. 1c. In this scenario, only an odd total number of mixing photons ($m + n =$ odd) was observed, and all even total number of harmonics disappeared, indicating the presence of inversion symmetry along the $\Gamma$-K direction.

To understand the influence of the inversion symmetry on the ultrahigh-order wave-mixing spectra, we performed numerical simulations to solve the semiconductor Bloch equations (SBEs) by incorporating the noncollinear two-pulse interaction at temporal overlap with an angle of $\theta \sim 18$ mrad and a beam size around $\mu \sim 50\ \mu m$ (see SI, Sec. 2). To directly compare with experimental results, we calculated the far field spatial-resolved spectrum $I(\omega, x)$ using experimentally measured parameters as input to the non-collinear numerical simulations of SBEs (see SI, Sec. 2.2). The simulated results, as shown in Fig. 1d, e, considering the inversion symmetry of the crystal (i.e., the complex transition dipole moment), demonstrate perfect consistency with the experimental results.

The optical properties of $\alpha$-quartz crystal are significantly influenced by the Coulomb interaction of electrons and holes[46]. This results in the existence of a fundamental exciton with a high binding energy of $E_{ex} = 1.2$ eV[23,46], which is self-trapped in the tetrahedral anion complex $(SiO_4)^{4-}$ (see inset in Supplementary Fig. 6). According to the ab initio study[47], the exciton localization length is up to 4.2 Å, which is comparable with the lattice constants, $a = b = 4.9$ Å and $c = 5.4$ Å[48], implying that the fundamental exciton in $\alpha$-quartz is of the Frenkel type[49]. Group-theoretical analysis of exciton energy spectrum gives the first excited and the second optically active states denoted as $1A_1$ and $2T_2$, respectively, where $A_1$ and $T_2$ are the irreducible representations of the tetrahedral symmetry group $T_d$[50].

Figure 2a illustrates the main mechanisms of HHG in solids, including intraband current densities $\tilde{J}_{i,\mathbf{K}}(\omega) \propto \mathcal{F}[n_{i,\mathbf{K}}(t)\nabla_{\mathbf{K}}E_{i,\mathbf{K}}(t)]$ depending on instantaneous band populations and group velocities, and interband polarization densities $\tilde{P}_{ij,\mathbf{K}}(\omega) \propto \mathcal{F}[\mathbf{d}_{ij,\mathbf{K}}\rho_{ij,\mathbf{K}}(t)]$ depending on the transition dipole matrix elements[18], where $\mathbf{K} \equiv$

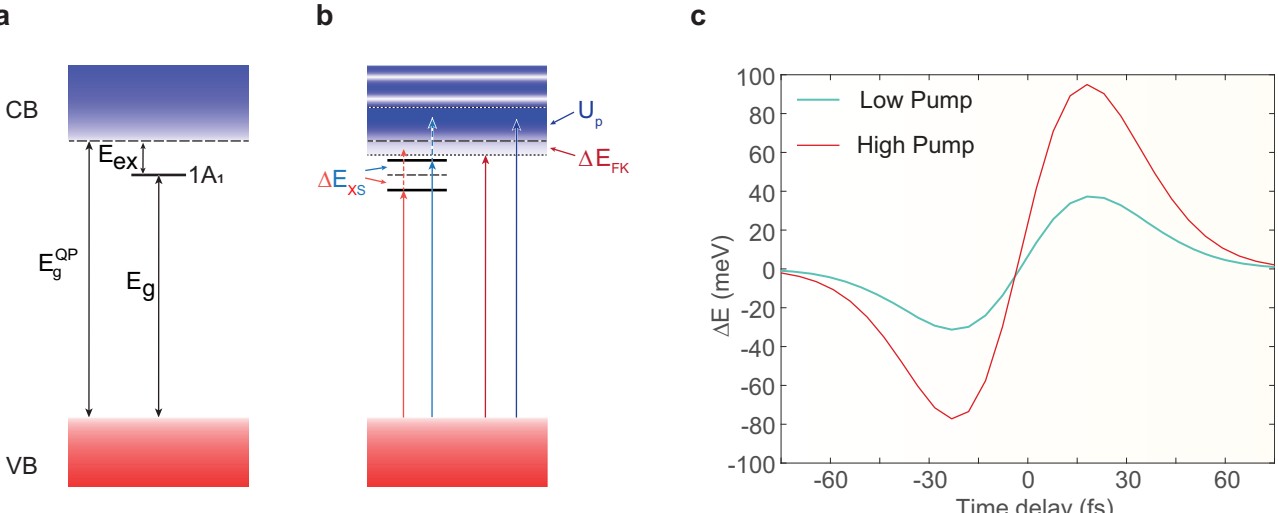

**Fig. 3 | Light-driven carrier energy shifts from self-energy perspective. a** Energy bands and the excitonic $1A_1$ state in the absence of an electric field. **b** Field-induced renormalization of excitonic and Bloch states. Charge carrier transitions in the presence of a strong, non-resonant electric field result in population-dependent excitonic energy shift ($\Delta E_{XS}$) due to the Stark effect, band edge shift ($\Delta E_{FK}$), and density of state oscillations due to dynamical Franz-Keldysh effect, as well as the ponderomotive effect (Up) within the continuum of Bloch states. The colors (red and blue) of the arrows correspond to the energy shifts. **c** An analytic result for the delay-dependent electron energy shifts derived from the dynamics described in the panel. In negative time delays, when the carriers are in the ground state, the redshift is caused by the excitonic AC Stark and dynamical Franz-Keldysh effects. For positive time delays, the field-induced virtual transitions between the excitonic and Bloch states as well as the ponderomotive effect, contribute to the excited carrier energies and induce a blueshift. The light blue and red lines represent the low and high pump conditions, respectively.

$K(t) = k + A(t)$ is the kinetic crystal momentum of carriers driven by the field, and $A(t) = -\int_{t_0}^{t} dt' F(t')$ is the vector potential. Hereafter, we omit the time argument of kinetic momentum, for brevity.

To investigate the dynamics of the coherent spectral response, a manifestation of AC Stark shift and dynamical Franz-Keldysh effect, we acquired integrated time-resolved harmonic spectra from the weaker probe field $\omega_2$ at the (0, 3) harmonic order. Figure 2b was acquired with the probe and pump pulse intensities set at $2 \times 10^{11}$ W/cm² and $4 \times 10^{12}$ W/cm², respectively. The trace exhibits a delay-dependent energy modulation of the spectrum profile, which varies as the delay $\Delta t$ is tuned from negative to positive values. By performing the centroid analysis of the spectra in the energy domain, represented by the dashed blue lines, we observe a modulation depth (peak-to-peak) reaching up to ~114 meV. This indicates a significant variation in the central energy of the spectral response as the delay $\Delta t$ is changed. We also captured the time-resolved spectra of the pump field, as depicted in Supplementary Fig. 2, where we observed an increase in the yields of the harmonics ($7^{th}$ to the $11^{th}$ order) without any significant centroid energy shift. This behavior contrasts with the energy-modulated spectral profiles observed in harmonics generated from the probe field. Recent work has highlighted the enhanced efficiency of HHG through a similar two-color field in silica[51]. The noted distinction between the harmonics produced by the weaker probe and the intense pump field underscores the field strength-dependent dynamics and responses of the system during the HHG process.

To understand the delay-dependent harmonic centroid shifts of the probe pulse, we performed quantum numerical simulations to solve the noncollinear SBEs in the length gauge using a two-band model. Figure 2c depicts the time-resolved data traces of the H3 signal from the probe pulse, simulated with similar probe and pump pulse intensities as those of the experimental data in Fig. 2b using a dephasing time $T_2 = 3$ fs. The numerical simulation successfully reproduces both the modulation of the harmonic centroid energy and the observed bleaching effect when the two pulses are close to overlapping. While higher-order contributions remain evident, they exhibit an opposite temporal evolution, resulting in a blueshift at negative delays and a redshift at positive delays. However, these effects are not experimentally observable due to the constrained signal-to-noise ratio of the measurements.

To confirm the excitation regime, we analyzed the pump intensity-dependent yields of the harmonics (Supplementary Fig. 3). The observed trend reveals that the power law fails to account for the behavior as pump intensity increases, suggesting the involvement of a non-perturbative regime in the current experimental settings. In a noncollinear geometry, the pure time-resolved HHG spectra of the probe can be interpreted as being dressed by the pump pulse, which modifies the electronic states and causes virtual transitions between them. This leads to significant modulations in the cycle-averaged energies and the yields of the HHG from the probe pulse. Therefore, the energy shift in the harmonic spectrum can be understood as the consequence of the perturbation of the Bloch and excitonic states by the charge carriers excited by the pump field. We found that under the influence of a pump pulse, the major contribution to the delay-dependent energy shift of exciton energy can be described within the second order of adiabatic perturbation theory (see SI, Sec. 3.1):

$$\Delta E_i^{(2)}(\Delta t) \approx \sum_{k, j \neq i} \left| \Omega_{ij, K_1}^{(0,1)}(\Delta t) \right|^2 \mathrm{Re} \left[ \frac{\Delta \bar{n}_{ij, K_1}(\Delta t)}{\Delta \bar{E}_{ij, K_1}(\Delta t) + i\gamma} \right] \quad (1)$$

Here, overline symbol means cycle averaging, indices $i$ and $j$ enumerate the excitonic states and electron-hole pairs in the Bloch bands, $K_1$ is the kinetic crystal momentum of the carriers in the pump field, $\Omega_{ij, K_1}^{(0,1)}(\Delta t) = F_{0,1}(\Delta t) \cdot d_{ij, K_1}(\Delta t)$ is the envelope Rabi energy, $F_{0,1}(\Delta t)$ is the pump pulse envelope multiplied by the unit vector in its polarization direction, $\Delta \bar{E}_{ij, K_1}(\Delta t) = \bar{E}_{i, K_1}(\Delta t) - \bar{E}_{j, K_1}(\Delta t)$ is the difference between the cycle-averaged instantaneous exciton energies, $\gamma = 1/T_2$ is the dephasing rate due to carrier-phonon and carrier-carrier interactions beyond the TDHF approximation, $\Delta \bar{n}_{ij, K_1}(\Delta t) = \bar{n}_{i, K_1}(\Delta t) - \bar{n}_{j, K_1}(\Delta t)$ is the difference between the cycle-averaged populations, which can be found by transforming the conduction ($c$) and valence ($v$) band populations in the electronic representation as

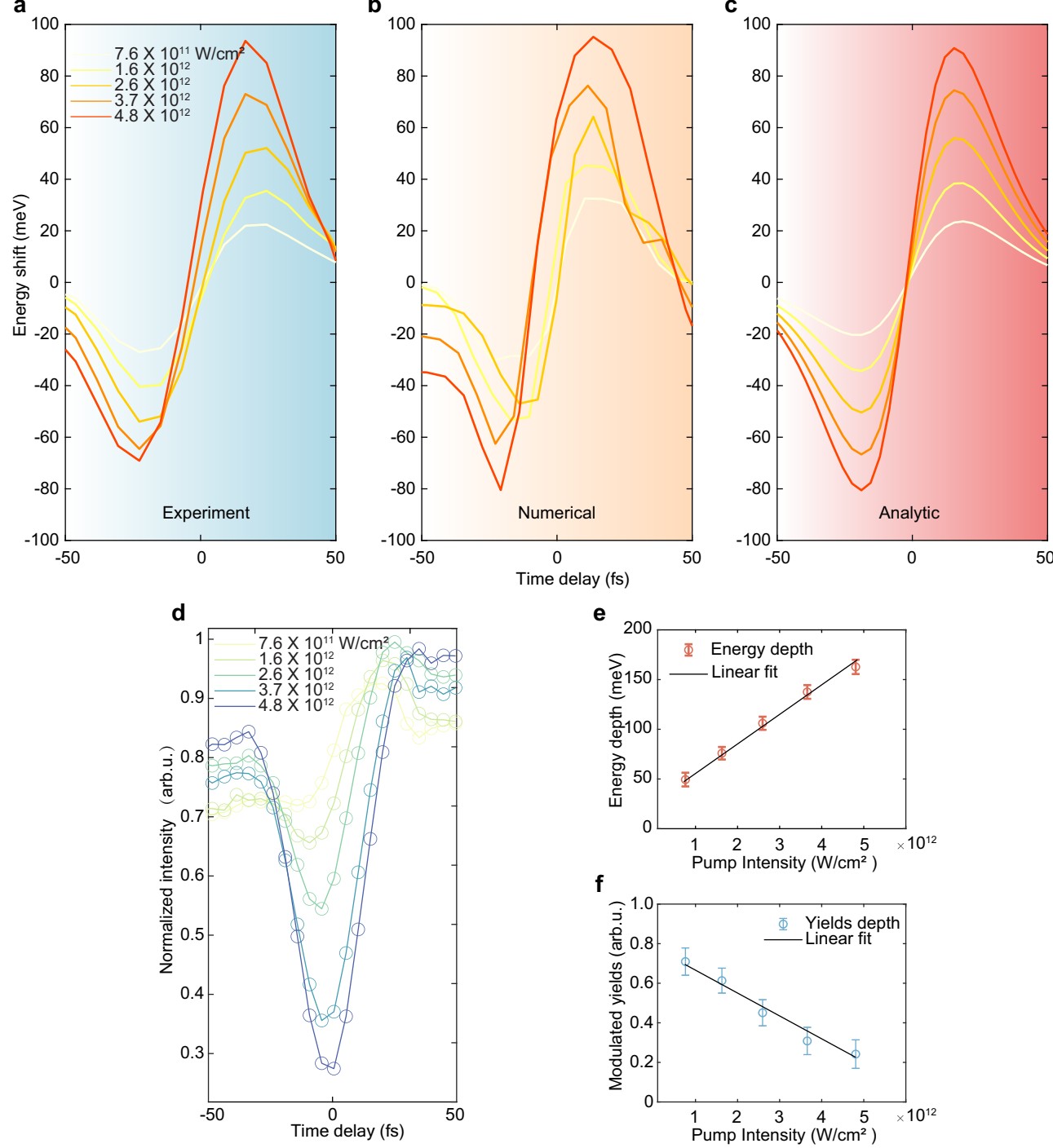

**Fig. 4 | Pump-pulse-dependent carrier energy shift. a** Integrated time-delayed central energy shift in the $H_3$ spectra of the probe obtained through intensity-scaling measurements when both pump and probe pulses are polarized along the Γ-M direction, respectively. The estimated peak intensity of the pump ranges from $7.6 \times 10^{11}$ to $4.8 \times 10^{12}$ W/cm², while the probe intensity is $2 \times 10^{11}$ W/cm². **b, c** Intensity-dependent numerical simulation and analytic solution when using similar parameters in the experimental setting. **d** Integrated time-delayed intensity modulations under different pump intensities. **e** Variation of the extracted energy modulation depth (peak-redshift to peak-blueshift) with the pump intensity, along with fits following the linear trend (solid black line). **f** Extracted modulation yields (spectral intensity) around zero delay under different pump intensities, with its linear fit shown as a solid black line. The error bars represent statistical uncertainties based on five measurements under identical experimental conditions.

$n_{i,\mathbf{K}_1}(\Delta t) = \sum_{c,v,\mathbf{k}'} \left| \mathcal{A}_{cv\mathbf{k}'}^{(i,\mathbf{K}_1)} \right|^2 n_{c,\mathbf{k}'}(\Delta t) [1 - n_{v,\mathbf{k}'}(\Delta t)]$, $\mathcal{A}_{cv\mathbf{k}}^{(i,\mathbf{K}_1)}$ are the coefficients of exciton state expansion obtained via solution of the Bethe-Salpeter equation, see Eqs. (3) and (4) in SI. $\mathbf{d}_{ij,\mathbf{K}_1}(\Delta t) = \langle i, \mathbf{K}_1 | i\nabla_{\mathbf{K}_1} | j, \mathbf{K}_1 \rangle$ is the transition matrix element. For excitonic states, it can be obtained via similar transformation of interband matrix elements.

We investigated the influence of the dephasing time $T_2$ on the energy shift, as shown in Supplementary Fig. 9. The centroid energy shift persists even at an extremely short dephasing time of 0.5 fs, which confirms its population dependence and robustness to decoherence. This may also offer a method for a measurement of the dephasing time $T_2$ in such a complex system. To better describe the dynamics of the

energy shifts in the presence of electron-electron interaction, we derived an analytic expression similar to Eq. (1). For the field-dressed excitonic states (see SI Section 3.2), the single-particle energies and optical matrix elements are replaced with the excitonic ones.

In the absence of an electric dressing field, the energy bands and the lowest excitonic $1A_1$ state are illustrated in Fig. 3a. As shown in Fig. 3b, before excitation of the charge carriers to excitonic and Bloch band states, the redshift appears due to excitonic Stark effect. After the excitation, the harmonics exhibit a blueshift due to the ponderomotive effect of carriers and virtual transitions from the lowest $1A_1$ state to higher excitonic or Bloch states. The instantaneous energy shift of the exciton, driven by the strong pump field, is given by the following expression:

$$\Delta E_{1A_1}^{(2)}(\Delta t) \approx \Delta E_{1A_1}^{(+)}(\Delta t) + \Delta E_{1A_1}^{(-)}(\Delta t) \tag{2}$$

Here, the first term

$$\Delta E_{1A_1}^{(+)}(\Delta t) = \int_{BZ} \frac{d^3k}{(2\pi)^3} \left\{ \left| \Omega_{1A_1\,2T_2,\mathbf{K}_1}^{(0,1)}(\Delta t) \right|^2 \mathrm{Re}\left[ \frac{\Delta \bar{n}_{1A_1\,2T_2,\mathbf{K}_1}(\Delta t)}{\Delta \bar{E}_{1A_1\,2T_2,\mathbf{K}_1}(\Delta t) + i\gamma} \right] \right.$$
$$\left. + \left| \Omega_{1A_1\,cv,\mathbf{K}_1}^{(0,1)}(\Delta t) \right|^2 \mathrm{Re}\left[ \frac{\Delta \bar{n}_{1A_1\,cv,\mathbf{K}_1}(\Delta t)}{\Delta \bar{E}_{1A_1\,cv,\mathbf{K}_1}(\Delta t) + i\gamma} \right] \right\} \tag{3}$$

describes the blue shift due to virtual transitions between the lowest $1A_1$, the next optically active exciton state $2T_2$ (first term in braces) and the Bloch states (second term in braces).

The second term

$$\Delta E_{1A_1}^{(-)}(\Delta t) = \int_{BZ} \frac{d^3\mathbf{k}}{(2\pi)^3} \left| \Omega_{1A_1\,0,\mathbf{K}_1}^{(0,1)}(\Delta t) \right|^2 \mathrm{Re}\left[ \frac{\Delta \bar{n}_{1A_1\,0,\mathbf{K}_1}(\Delta t)}{\Delta \bar{E}_{1A_1\,0,\mathbf{K}_1}(\Delta t) + i\gamma} \right] \tag{4}$$

accounts for the red shift in the delay-dependent H3 spectra due to the excitonic AC Stark effect, which dominates at negative delays, when the charge carriers are primarily in the ground state.

According to the Eq. (2), the results obtained, as illustrated in Fig. 3c, qualitatively reproduces the features of both the experimental data and numerical calculations, except the negative time delays with an increase of intensity. As before the time overlaps, i.e., $\Delta t < 0$, the pump dressing field leads to a dominance of excitonic states. On the other hand, a similar crossover of strong-field effects exists on the single-particle level with band states. There is the dynamical Franz–Keldysh effect reducing the band-gap and modifying the density of states (see Fig. 3b). When $\Delta t > 0$, a significant number of carriers can be excited to the $1A_1$ excitonic state and to the conduction band, by exciton dissociation or via direct interband transitions.

The charge carriers excited to the Bloch bands could receive further energy from the laser field via the ponderomotive effect, which results in a blue shift given by expression[2] generalized to the TDHF approximation:

$$\widehat{U}_{p,\mathbf{k}}(t) = \frac{1}{T_{0,1}} \int_{t-T_{0,1}/2}^{t+T_{0,1}/2} \Delta \widehat{E}_{cv,\mathbf{K}_1}(t_1)\, dt_1 - \Delta \widehat{E}_{cv,\mathbf{k}}, \tag{5}$$

where energies of electron-hole pairs are renormalized by the Coulomb interaction

$$\Delta \widehat{E}_{cv,\mathbf{K}_1}(t_1) = \widehat{E}_{c,\mathbf{K}_1}(t_1) - \widehat{E}_{v,\mathbf{K}_1}(t_1), \quad \widehat{E}_{i,\mathbf{K}_1}(t_1) = E_{i,\mathbf{K}_1} - \sum_{\mathbf{q} \neq \mathbf{K}_1} V_{|\mathbf{K}_1 - \mathbf{q}|}\, n_{i,\mathbf{q}}(t) \tag{6}$$

To quantitatively assess the impact of energy shifts, we conducted experiments with the intensity of the probe pulse held constant at $2 \times 10^{11}\,W/cm^2$. We recorded the time-resolved harmonic spectra while varying the pump intensities in the range of approximately $7.6 \times 10^{11}$ to $4.8 \times 10^{12}\,W/cm^2$. Both the pump and probe pulses were polarized along the Γ-M direction.

As shown in Fig. 4a, the integrated energy shifts exhibit significant modulations in the time-delayed H3 spectra of the probe, as observed through power-scaling measurements. These results align well with the quantum simulations (Fig. 4b) and analytical results (Fig. 4c) based on the calculated delay-dependent electron density variations. We observed that, under moderate pump amplitudes, the blue and red shifts of the harmonic centroid energy exhibit symmetric growth. However, as the pump intensity increases, the centroid energy shift displays a squeezed asymmetric profile, with the blue shift being larger than the red shift. This anisotropy can be understood as a consequence of asymmetry between the coefficients of perturbative expansion for transitions from the ground state and the first excited state. They are given by the ratios of the squared modulus of the transition matrix element divided by the energy difference of the virtual transition. The higher the pump field intensity, the more significant the asymmetry between the red and blue shifts. Additionally, subtle temporal behavior is observed in the analytical spectra, which is symmetric about zero-time delay. In contrast, the harmonic centroid shift reverses negative delays in the other two cases, a discrepancy that can be attributed to higher-order effects.

The modulation intensity depth, represented by the peak-to-peak variation after intensity normalization, is extracted and presented in Fig. 4d. The experimental results agree well with our model as shown in Fig. 4e, which employs the fitting model $\Delta E \propto I_1$, indicating the feasibility of achieving continuous manipulation of electron energy by varying the instantaneous driving field intensity. Considering that the influence of the pump field operates within the nonperturbative regime, the observed bleaching effect demonstrates a linear scaling, as depicted in Fig. 4f. Our findings establish a foundation for dynamically tuning carrier energies on fs timescales, which can be applied in spectral-based all-optical switches and sensing within quantum systems.

In summary, we demonstrate that noncollinear harmonic spectroscopy offers a powerful and versatile platform for dynamically controlling carrier motion via tailored light fields. By unraveling the interplay among key strong-field phenomena—the AC Stark effect, dynamical Franz–Keldysh, field-induced coherent coupling of exciton and Bloch states, and ponderomotive effect—we establish a robust framework for studying the dynamics of carrier energies and many-body effects to explore how variations in external conditions influence these interactions. The ability to induce redshifts and blueshifts of carrier energies within a single pump-probe scan highlights the essential role of both adiabatic and nonadiabatic processes in the driven and open quantum system. This experimental approach can be generalized to other systems, such as wide-bandgap two-dimensional (2D) materials, such as nitrides[52], transition metal dichalcogenides (TMDs)[53], which exhibit optical properties similar to those of α-quartz crystal and are characterized by strong excitonic effects. HHG has been successfully demonstrated in 2D materials[16,17], making such experiments feasible for further exploration of electron-electron correlations. Due to a smaller band gap, one can conduct these experiments using mid-infrared or THz pulses. Noncollinear harmonic spectroscopy deepens our understanding of light-induced phenomena in quantum materials and lays the groundwork for engineering next-generation optoelectronic and nanophotonic devices. By harnessing strong light-matter interactions, our work opens new frontiers in exploring non-equilibrium phases of matter and designing quantum materials with bespoke functionalities, advancing the broader fields of ultrafast science, modern photonics, and quantum technology.

## Methods

### Sample

In our experiments, we employed a z-cut $SiO_2$ (α-quartz) crystal ([0001] orientation) provided by United Crystal, with dimensions of $5 \times 5$ mm, as the target for high-order harmonic generation. The crystal surface was optically polished on both sides, ensuring high-quality optical properties. To determine the crystal thickness, we utilized a custom-made white light interferometry spectrometer, which measured the thickness to be $20 \pm 5$ μm.

### Experimental setup

As shown in Fig. 1a and Supplementary Fig. 1a, we experimented with an intense fundamental laser pulse ($\omega_1 = 800$ nm) intersecting a weaker second-harmonic field ($\omega_2 = 400$ nm) non-collinearly with the sample. The fundamental and second harmonic were recombined using a dichroic mirror, with beams spaced vertically by 5 mm. A Ti: Sapphire amplifier generated the pump pulses at 800 nm, with a repetition rate of 10 kHz and total energy of 1 mJ, maintaining P-polarization. The sample was aligned precisely using a rotational stage to optimize interaction with the laser pulses. Pulse durations were measured via our TG-FROG setup, yielding $25 \pm 3$ fs (Supplementary Fig. 1b, c) and $28 \pm 4$ fs (Supplementary Fig. 1d, e). A 25 cm focal length lens concentrated the light pulses to a size of approximately 25–40 μm, with peak intensities ranging from ~$10^{11}$ to ~$10^{13}$ W/cm². HHG spectra were captured using an EUV spectrometer covering 6 eV to 35 eV, with a resolution of ~0.1 eV. The non-collinear angle was estimated at ~18 mrad, and a piezo stage was used for precise delay scans. For further details, see the SI, Section 1.

### Numerical simulation

To simulate the non-equilibrium dynamics, we employ a non-collinear 1D numerical simulation of the time-dependent SBEs that incorporates electron-electron interactions using the Hartree-Fock approximation with a soft-Coulomb potential. Further details are provided in the SI, Section 2.

## Data availability

The data supporting the findings of this study and its Supplementary Information are available upon request.

## Code availability

The codes that support the findings of this study are available upon request.

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

## Acknowledgements

It is our pleasure to acknowledge the fruitful discussion with Tilmann Kuhn. Tran Trung Luu gratefully acknowledges funding support by the GRF project 17315722, the NSFC/RGC Joint Research Scheme N_HKU7104/22, and the Area of Excellence project AoE/P-701/20. Liang-You Peng acknowledges the support by the National Key Research and Development Program of China (Grant No. 2024YFA1612101) and the National Natural Science Foundation of China (Grants No. 12234002 and No. 92250303).

## Author contributions

J.Z. and T.T.L. conceived and designed the experiment; J.Z., T.D.T., W.Y., C.Z., Z.W., and W.X. contributed to building the experimental set-up; J.Z. conducted the experiments and performed the data analysis. S.Yu.K., L.Y.P., X.L., L.G., and J.N.Z. developed the theoretical model; J.Z., W.X., X.L., and T.D.T. performed numerical simulations with SBEs; S.Yu.K. performed numerical simulations of material properties with the VASP code using DFT, $GW_O$-method, and solution of the Bethe-Salpeter equation (BSE), as well as analytic derivations of non-equilibrium dynamics with the density matrix formalism. J.Z. and S.Yu.K. drafted the initial manuscript. All authors contributed to the discussion and interpretation of the experimental data. T.T.L. supervised the project.

## Competing interests

The authors declare no competing interests.
