## [Transparent Peer Review file · Nature Communications]

Noncollinear Harmonic Spectroscopy Reveals Crossover of Strong-Field Effects

Corresponding Author: Professor Tran Trung Luu

Version 0:

Reviewer comments:

Reviewer #1

(Remarks to the Author)

The manuscript entitled Channel-resolved femtosecond switching of carrier energies by Zhang et al. demonstrates the initiation, monitoring and control of the femtosecond dynamics of carriers within the non-collinear high harmonic spectroscopy of the energy bands. The field of ultrafast light matter in solids is booming, with several promising techniques for observing and controlling material properties on the timescale of laser pulses. These techniques not only advance our understanding of quantum systems, but also facilitate the engineering of carrier dynamics in advanced devices and nanophotonics for next-generation electronic devices.

However, many questions remain unanswered. These are mainly due to the lack of well-established probing tools. Here, the authors use non-collinear high harmonic spectroscopy as a tool to probe femtosecond dynamics in solids. The author uses their experimental data, combined with numerical and analytical analysis, to comprehensively study the dynamic energy shifts of Bloch states and excitonic levels. Using their spectroscopy tool, several results are reported: (1) red shifts of harmonics due to excitonic strong effects, (2) dynamic Franz-Keldysh effects, (3) Bloch states, (4) ponderomotive effects, (5) even order harmonics for asymmetric fields, and (6) exciton assembly and dissociation. Using their experimental capabilities and model simulations, they present these phenomena, disentangle them, and use model simulations to explain them. The manuscript is up to date and the results are well presented and understandable to experts in the field. I have only a few questions and suggestions that should be addressed before I can recommend publication.

Non-collinear non-linear mixing, as used in sum-frequency or difference-frequency generation, is not new and has been used as a spectroscopic tool in various systems. Here the author focuses on the high harmonic analogue, a well established technique. In particular, the demonstration that this technique is suitable for tracking various ultrafast processes in solids is a good reason why this manuscript should be published. While I understand that there are some length restrictions for a wider audience, including some in nonlinear optics, it would be good to discuss why non-collinear high harmonic spectroscopy is more suitable than conventional HHG. This is of course obvious to someone working in the field, but not to a general reader. Along this line, different measurements are made to extract different phenomena (see (1)-(6)). So I wonder about the clear message of the manuscript. Are they investigating a new effect or a new method? As it is currently written, it seems that new effects are being observed, also given the title "Channel-Resolved Femtosecond Switching of Carrier Energies". However, most of the effects presented have been studied in solids, i.e. AC Stark shifts, Keldysh effects, etc. If the author would rather present a new method that is sensitive to all of these effects, I would suggest that the author change the title to focus more on the method and then discuss why this method is more suitable for studying the phenomena presented compared to other effects.

Finally, do I understand correctly that a dephasing time of 0.5 fs is used in their SBE model simulation? Why is it so much shorter than that used in other SBE simulations?

(Remarks on code availability)

Reviewer #2

(Remarks to the Author)

Peer Review Report: Channel-Resolved Femtosecond Switching of Carrier Energies, Jicai Zhang et al.

Zhang and coauthors present a comprehensive experimental and numerical investigation of strong-field-induced electron and exciton dynamics in the well-known insulator SiO₂—a material extensively studied within the strong-field community. Over the past decade, the optical response of SiO₂ under intense, ultrashort 800 nm laser fields has been widely explored using various experimental techniques, achieving temporal resolutions down to the sub-femtosecond scale. In this work, however, the authors introduce a novel approach based on high-harmonic spectroscopy, applied while the sample is driven far from equilibrium by an intense, non-resonant pump field in a non-collinear geometry. This technique, previously demonstrated primarily in gas-phase systems, uniquely combines the capability of high-harmonic generation (HHG) to probe conduction-band electron dynamics with the coherent manipulation of the crystal's harmonic response via a delay-controlled strong-field pump pulse. The resulting data provide an original and compelling perspective on the intricate interplay between light, electrons, and excitons—all contributing to the optical response of SiO₂ in the UV–EUV spectral range.

The manuscript also includes numerical simulations based on semiconductor Bloch equations, as well as a simplified analytical model, both of which support the experimental findings. These theoretical tools successfully reproduce the observed shift in the center-of-mass of the third-harmonic signal (H3) generated by the 400 nm probe pulse across different pump laser intensities. This allows the authors to disentangle the contributions of various physical effects—including the AC Stark shift and the dynamical Franz–Keldysh effect—to the complex light-driven dynamics observed.

I find the proposed methodology highly interesting and complementary to existing experimental techniques. The quality of the data—both experimental and numerical—is excellent and convincingly demonstrates the potential of this approach to investigate a broad range of light-induced phenomena. However, the authors concentrate on the analysis of a single harmonic peak (H3), somehow neglecting all the other signals, while still presenting the technique as ultrahigh-order wave mixings without providing a clear explanation. In my opinion, all the datasets and their analysis could be presented with a higher level of detail: it is this lack of clarity, in my opinion, that raises some concerns on the physical interpretation provided by the authors. Finally, I am also not totally convinced by the authors regarding the possibility of extending this approach to any other quantum material. Please see the attached comments, divided into major and minor points, to better understand what I mean.

In conclusion, I believe the study is original, significant, and holds potential for publication in Nature Communications. Nevertheless, I would like the authors to address all the points listed below and provide a revised version of the manuscript before I can recommend their work for publication.

MAJOR

- Title and abstract: In my opinion, the terms “Channel-resolved” and “Switching of carrier energies” are not well-established or clearly defined concepts and may not be easily understood by non-expert readers. The title may remain as it is, but a clearer explanation should be included in the abstract. Which channels are referred to? What does “switching” mean in this context? On first reading, I interpreted it as a binary ON/OFF process, while the data presented here rather indicate a manipulation—a continuous fine-tuning—of the excited carriers' energy.
- Introduction, page 3 line 5: Why do the authors refer to the “electronic wavepacket center-of-mass” when discussing the temporal evolution of the H3 signal from the probe pulse? While this is certainly related to electron dynamics, the phrase may suggest that an electron measurement was performed, rather than a photon-based one. To avoid confusion, I suggest rephrasing.
- Page 5, line 12: This is the first mathematical formula in the text. The authors should define and explain every physical quantity introduced.
- When the H3 spectrum is discussed for the first time (page 5), the authors state that the trace exhibits an “asymmetrical spectrum profile.” However, the data in Fig. 2 do not clearly show this asymmetry. The spectrum at large negative delays appears rather symmetrical. Is this asymmetry significant? If so, the authors should present data that clearly illustrate this and explain the underlying physical cause.
- Page 5, lines 24-28: The authors briefly comment on the differences between the time-resolved harmonic spectra of the pump field and the H3 signal of the probe. I believe this is a key point in the manuscript and deserves more detailed discussion. Is the observed difference due to the fact that, in the case of the H3 signal from the probe, strong-field pumping is involved, while in the other case (switching pump and probe pulses) the actual pump is less intense? Also, intensity values used in Supplementary Fig. 2 are not reported. Alternatively, could the H3 signal be more sensitive to the conduction band minimum and excitonic response—regions where most pump-induced phenomena occur—while the pump harmonics are not? In my view, this point is crucial for understanding the manuscript and should be significantly expanded.
- Page 6, lines 6-9: Are the authors referring to the numerical trace in Fig. 2c (rather than 2b, as written)? If so, the term “acquired” is misleading and should be replaced. Otherwise, the discussion becomes confusing, since Fig. 2b was already introduced on the previous page. Additionally, the authors should explain why the pulse intensity values reported here differ from the initial values provided, especially for the probe pulse.
- Page 6, line 9: The authors mention an “observed bleaching effect” at temporal overlap, apparently reproduced by the numerical data. However, when comparing Figs. 2b and 2c, the bleaching is much more pronounced in the numerical trace. Moreover, a secondary contribution is evident, showing an opposite temporal evolution (blueshift at negative delays, redshift at positive ones). Can the authors clarify this? A more thorough discussion of the bleaching effect should be included.
- Page 6, line 20: The perturbation of Bloch and excitonic states is attributed to “charge carriers excited by the pump field.” What kind of excitation is this? Are real carriers excited via photon absorption, or is this a virtual process? What are the initial and final states involved? Given SiO₂'s large band gap, a real excitation would require absorption of at least six photons—an inherently low-probability process. This seems inconsistent with the field being described as non-resonant (see Fig. 3 caption).
- Page 6, line 21: After attributing the effect to charge carriers, the authors introduce a model based on the independent particle approximation, which seems contradictory. Is this approximation justified in this context? The authors should clarify this point.
- Page 8, lines 9-16: The discussion of the analytical model lacks clarity. It would be very helpful to plot the temporal

evolution of both terms (Eqs. 3 and 4), along with their sum and the pump pulse profile. Also, the statement “when $\Delta t > 0$, the field strength increases” is unclear—isn’t the pump field temporally symmetric? The authors also state that “a significant number of electrons are excited to conduction bands.” Can they provide an estimate? Lastly, processes such as “exciton ionization” and “interband tunneling of electrons” should be described in greater detail, considering their importance for interpreting the system’s complex optical response under strong fields.

- Page 11, lines 26-28: This final section is very interesting, but the comparison between a bulk SiO₂ crystal and a 2D material seems questionable. In a 2D material, the substrate indeed strongly affects the optical response, while in a bulk crystal, its influence on the volume-averaged THG signal should be far less significant. The comparison is not well justified.
- The authors claim that their framework is general and applicable to quantum materials. I would temper this statement. While the work is certainly valuable and insightful, its applicability to materials beyond wide-gap insulators is not obvious. SiO₂, with its clear excitonic resonance and large band gap, allows observation of specific nonlinear effects that may not appear—or may manifest very differently—in materials with smaller band gaps, such as TMDs. In such systems, the balance between excited carriers and excitons could shift dramatically, altering the underlying physics. This should be acknowledged more cautiously.

MINOR

- Abstract: The phrase “carrier-carrier interactions” is cited as part of “the intricate interplay between light and excited carriers.” However, light is not directly involved in carrier-carrier interactions. I suggest separating light-carrier from carrier-carrier interactions for clarity.
- Abstract: The material under investigation (SiO₂) is only mentioned at the end of the abstract (line 31). This should be stated earlier (e.g., at line 22) to avoid suggesting that the findings apply universally.
- Introduction, line 2: Remove “The” before “light-matter interaction”.
- Introduction, lines 4-6: The sentence implies that few-cycle pulses can only be generated via light pulse synthesis, which is inaccurate. Common methods (e.g., hollow-core fibers and chirped mirrors) also apply. Also, given the 10–20 cycle pulses used in this study, is this even relevant? Consider rephrasing.
- References: I suggest removing Ref. 25 unless there’s a citable arXiv version of the manuscript.
- Page 2, lines 29–33: I recommend citing the work of Sidiropoulos et al., Nat Commun 14, 7407 (2023), which is highly relevant to the discussion of many-body effects induced by strong light fields.
- A period is missing at the end of the Introduction.
- Figure 1: While the technique enables temporal manipulation of harmonic emission via a pump pulse, the presented datasets are not temporally resolved. The authors should state in the caption that the pump-probe delay is set to 0. Also, correct “upwards” to “downwards” in line 5, if that’s what the sketch suggests. Additional typos: “it’s” and “fields”.
- Page 5, line 30: Typo in “noncolinear”.
- Page 7, Equation 1: The term “ $F_{0,1}(\Delta t)$ ” should be explained just like the others.
- Page 8, line 16: This sentence repeats what was written two lines earlier—consider removing or rephrasing.
- Page 9, lines 19–20: The authors note good agreement between experiment, simulation, and analytical models in the power-scaling results. While the amplitudes align, there is a subtle but interesting difference in the temporal behavior. The analytical trace is symmetric about time zero, while in the other two cases, the COM shift reverses for negative delays. This discrepancy might point to underlying physical effects not captured by the analytical model and deserves a brief comment.
- Page 10, line 11: It is unclear what maximum was used for normalizing the curves, as none reach 1. Please clarify.
- Page 11, lines 4–9: The method used to extract the three quantities shown in Fig. 4 d–f is unclear. Consider explaining it in more detail or providing a supporting graph in the Supplementary Information.
- Page 11, line 26: “Atomically thin” and “two-dimensional” sound redundant—consider removing one.
- SI, page 11, Fig. 5 caption: Are the authors plotting static absorption, as written in the caption, or optical conductivity, as labeled on the graph? Also, specify whether it is the real or imaginary part.
- SI, page 12, line 205: “Normalized to maximize...” what? The sentence is incomplete.
- SI, page 15, line 269: Are the assumptions “charge carriers only excited by the pump pulse” and “the probe pulse is short” justified if the probe pulse can generate harmonics and has a pulse duration of 28 fs (~21 optical cycles)? Please comment on this.
- SI, page 18, line 334: “Reference not found”—please correct.

(Remarks on code availability)

Although no usable code is provided, I am not an expert in numerical simulations, but I believe the reference provided by the authors and the supplementary material should contain all the information needed to reproduce the numerical results reported in the manuscript.

Reviewer #3

(Remarks to the Author)

The present manuscript by Zhang et al. reports on an experimental study of non collinear high harmonic generation, the purpose of which was to probe temporal carrier dynamics in an alpha-quartz crystal. The authors investigated the effects on the third harmonic (H3) energy of the probe beam (400 nm) as a function of the delay between pump (800 nm) and probe beams. Their observations included a red shift of the H3 signal for negative time delays, followed by a blue shift at positive time delays. The authors reproduce this experimental observation through two means: firstly, through numerical simulations (SBE); secondly, through an analytical derivation (adiabatic perturbation theory). Furthermore, the authors observe a bleaching effect on the H3 signal when the two beams are temporally overlapped (time delay equal to zero). The study goes on to consider the dependencies of these two effects as a function of the pump power intensity, claiming the possibility to manipulate the electron energies by the driving field intensity. The report also presents a simulated dependence of the CoM depth on the electron-hole Coulomb interaction potential, which the authors claim to be useful in the study of many-body

effects in solid.

I think the work has some interesting parts, like the maths behind eq. 2-3-4 in the main text or the reported experimental data, but there are some things that need to be fixed before it can be published. My main concern is about the chirp of the two beams. How do the authors make sure that the beam's chirp isn't affecting what they're seeing? Do they just look at the time-resolved HHG for the pump pulse? My second concern is about a paper recently published in Nat Com that reports very similar studies (<https://doi.org/10.1038/s41467-024-52774-9>). Could the author explain what they are doing differently compared to that paper?

There are a few minor issues that need to be addressed:

-On page 3, line 19, where is the dashed line?

-In figure 1, it would be useful to report the same y-scale for the simulated and measured contour plots. What information can be extracted from the intensity dependence of the HH signals? Lastly, there is a typo in panel c at 14 eV: there should be (1,4) instead of (2,4).

-L. 18 p. 5, it is not specified the crystal direction where the H3 signal of figure 2 was acquired. When comparing to the result in the SI, like fig. 2, it should be important that the direction is the same (Γ -K). Also, have you tried the same experiment but in the other direction (from Γ to M)?

-Fig. 2, panel a: the y-scale doesn't have labels, and the numerical simulation was calculated using pulse intensities that are different from the experimental ones (9 vs. 0.2 PW/cm² for the probe). What does the simulation show with the experimental values?

-In equation 1, please explain what the subscript 1 of K means.

-In figure 3, panel b, please explain the meaning of the colour difference between the arrows.

-In figure 4, the error bars in panels e and f do not go along the x-axis, so they cannot be related to fluence uncertainties.

-The inset in Fig. 5 is not clear to me, and it should be stated in the caption that this result is due to a simulation and not to an experiment.

-Lastly, I wonder if the dephasing time T2 could be extracted by the experimental traces.

-In the references related to Science there is always the year 1979, I believe it is a typo.

(Remarks on code availability)

Version 1:

Reviewer comments:

Reviewer #1

(Remarks to the Author)

The reviewer has addressed my comments. In particular, I like the new title, abstract, and conclusion. I am also excited to see that the authors have further improved their manuscript by taking the other reviewers' comments into account.

(Remarks on code availability)

Reviewer #2

(Remarks to the Author)

I want to thank the authors for their significant efforts in addressing all the comments and questions raised by the three reviewers.

I firmly believe that the manuscript has improved in clarity and readability. Experimental and simulated datasets are now better presented, and their interpretation is more solid and convincing. Also the Conclusion and Outlook section results to be more precise about future applications of this technique to the study of more complex systems.

In conclusion, I am happy to recommend the actual version of the work for publication in Nature Communications.

(Remarks on code availability)

Reviewer #3

(Remarks to the Author)

In the revised version of the paper, Zhang et al. provided detailed and reasonable responses to all the queries raised by the other reviewers.

Regarding my personal comments, I am satisfied with the author's responses. Therefore, I believe that the revised version of the paper meets publication criteria. However, I would suggest improving panel b of Fig. 3 by adding in the caption what the authors mean by 'Up', 'DeltaExs' and 'DeltaEFK' to improve readability.

(Remarks on code availability)

Response to the Referees

First and foremost, we would like to express our sincere appreciation to the Referees for the time and effort they dedicated to reviewing our work. We are grateful for their thoughtful and constructive comments and questions.

We have conducted extensive data analyses and simulations to fully address the Referees' concerns and incorporate their feedback into our revised manuscript. We have significantly revised the full text, including title, abstract, main text, conclusion and outlook. Additionally, we have created new Supplementary Figures 4 and 5, revised main Figures 1-2, 4, and Supplementary Fig. 10. The results of these efforts are presented in the revised manuscript and this response to the Referees.

Finally, the invaluable expertise of the Referees has significantly strengthened our work. We firmly believe that our revised version represents a substantial improvement, and we hope it will be favorably considered for publication in *Nature Communications*.

To ease the Referee's reading, **their original texts are denoted in blue**, **our responses are in black**, and the **additions to the manuscript are in green**. **The corrected texts in the revised manuscript are marked in red.**

Referee #1

“The manuscript entitled Channel-resolved femtosecond switching of carrier energies by Zhang et al. demonstrates the initiation, monitoring and control of the femtosecond dynamics of carriers within the non-collinear high harmonic spectroscopy of the energy bands. The field of ultrafast light matter in solids is booming, with several promising techniques for observing and controlling material properties on the timescale of laser pulses. These techniques not only advance our understanding of quantum systems, but also facilitate the engineering of carrier dynamics in advanced devices and nanophotonics for next-generation electronic devices. However, many questions remain unanswered. These are mainly due to the lack of well-established probing tools. Here, the authors use non-collinear high harmonic spectroscopy as a tool to probe femtosecond dynamics in solids. The author uses their experimental data, combined with numerical and analytical analysis, to comprehensively study the dynamic energy shifts of Bloch states and excitonic levels. Using their spectroscopy tool, several results are reported: (1) red shifts of harmonics due to excitonic strong effects, (2) dynamic Franz-Keldysh effects, (3) Bloch states, (4) ponderomotive effects, (5) even order harmonics for asymmetric fields, and (6) exciton assembly and dissociation. Using their experimental capabilities and model simulations, they present these phenomena, disentangle them, and use model simulations to explain them. The manuscript is up to date and the results are well presented and understandable to experts in the field. I have only a few questions and suggestions that should be addressed before I can recommend publication.”

Our Response: We sincerely thank the Referee for the thorough evaluation of our manuscript and the encouraging feedback regarding the relevance and clarity of our work in advancing ultrafast light-matter interaction studies. We also appreciate the Referee's recognition of our use of non-collinear high-harmonic spectroscopy to probe femtosecond dynamics in solids and the insights into the significance of our findings. Following, we are fully committed to addressing the suggestions in detail and refining the manuscript accordingly. Thank you again for your thoughtful review.

Specific comments from the Referee

1. “Non-collinear non-linear mixing, as used in sum-frequency or difference-frequency generation, is not new and has been used as a spectroscopic tool in various systems. Here the author focuses on the high harmonic analogue, a well established technique. In particular, the demonstration that this technique is suitable for tracking various ultrafast processes in solids is a good reason why this manuscript should be published. While I understand that there are some length restrictions for a wider audience, including some in nonlinear optics, it would be good to discuss why non-collinear high harmonic spectroscopy is more suitable than conventional HHG. This is of course obvious to someone working in the field, but not to a general reader.”

Our Response: We thank the Referee for the insightful feedback regarding the application of the non-collinear nonlinear spectroscopy technique in our manuscript. We agree that this technique has been

applied in many systems. For example, in the gas phase case, the author has demonstrated that even high harmonic generation is a highly nonperturbative process, while a perturbation theory can be developed around it [1]. As mentioned by Referee #3, it has also recently been applied to amorphous silica, where the authors demonstrated that it enhanced the efficiency of high-order harmonics using a two-color non-collinear wave mixing setup [2]. We agree with the Referee that we should enhance the discussion in our manuscript to clarify why non-collinear high-harmonic spectroscopy offers distinct advantages over conventional collinear HHG settings.

In the revised manuscript, we added one sentence and related reference to discuss that the noncollinear geometry is suitable than conventional collinear HHG measurement: “A similar experimental arrangement has been previously discussed in the context of ultrahigh wave-mixing measurements in the gas phase⁴⁶. Compared to conventional collinear pump-probe geometry, this noncollinear configuration offers several advantages, including back-ground-free measurements by avoiding two-photon sideband absorption and energy overlap under two-color field conditions, as well as the capability for momentum- and parity-resolved high-harmonic generation (HHG) spectral measurements.”

2. “Along this line, different measurements are made to extract different phenomena (see (1)-(6)). So I wonder about the clear message of the manuscript. Are they investigating a new effect or a new method? As it is currently written, it seems that new effects are being observed, also given the title “Channel-Resolved Femtosecond Switching of Carrier Energies”. However, most of the effects presented have been studied in solids, i.e. AC Stark shifts, Keldysh effects, etc. If the author would rather present a new method that is sensitive to all of these effects, I would suggest that the author change the title to focus more on the method and then discuss why this method is more suitable for studying the phenomena presented compared to other effects.”

Our Response: We thank the Referee for making a very good point, which allowed us to clarify the message of our manuscript. In the revised version, we focused on the effect of crossover between the red shifts due to excitonic AC Stark effect and dynamic Franz-Keldysh effect, and the blue shifts due to field-induced virtual transitions between excitonic and Bloch states, and ponderomotive effect. To address the comment of Referee #1 and the related comment 1 of Referee #2, we changed the title and introduced the changes to the abstract and text of the manuscript.

In the revised manuscript, we rephrased the title to: “**Noncollinear Harmonic Spectroscopy Reveals Crossover of Strong-Field Effects**” and introduced the following changes to the abstract and summary of the manuscript.

New Abstract:

“**The ability to control electron motion with light fields represents a transformative frontier in modern physics, enabling dynamic manipulation of material properties at ultrafast timescales. Yet, the complex interplay between light and excited carriers—via mechanisms such as the AC Stark effect, field-induced**

coupling of excitonic and Bloch states, the dynamical Franz-Keldysh effect, and the ponderomotive effect—continues to challenge our understanding of quantum systems driven far from equilibrium. Here, we establish non-collinear high harmonic spectroscopy as a powerful technique for initiating, tracking, and steering femtosecond carrier dynamics across the energy landscape in the dielectric SiO₂ crystal. We achieved precise control over carrier energies, observing redshifts associated with the AC Stark effect in excitonic states and the dynamical Franz-Keldysh effect in Bloch states, alongside blueshifts induced by the ponderomotive effect and virtual transitions between the excitonic and Bloch states. Combining the rigorous numerical simulations with analytical theory, we identify the main mechanisms responsible for the crossover of different strong-field phenomena, responsible for the delay-dependent energy shift of excitonic or Bloch states. This control over the electronic and excitonic states opens new opportunities for tailoring carrier dynamics in quantum materials, paving the way for next-generation optoelectronic and nanophotonic technologies.”.

New Conclusions and Outlook section:

“In summary, we demonstrate that noncollinear high-harmonic spectroscopy offers a powerful and versatile platform for dynamically controlling carrier motion via tailored light fields. By unravelling the interplay among key strong-field phenomena—the AC Stark effect, dynamical Franz-Keldysh, field-induced coherent coupling of exciton and Bloch states, and ponderomotive effect—we establish a robust framework for studying the dynamics of electron energies and many-body effects to explore how variations in external conditions influence these interactions. The ability to induce redshifts and blueshifts of carrier energies within a single pump-probe scan highlights the essential role of both adiabatic and nonadiabatic processes in the driven and open quantum system. This experimental approach can be generalized to other systems, such as wide-bandgap two-dimensional (2D) materials, such as nitrides⁵², transition metal dichalcogenides (TMDs)⁵³, which exhibit optical properties similar to those of α -quartz crystal and are characterized by strong excitonic effects. HHG has been successfully demonstrated in 2D materials^{16,17}, making such experiments feasible for further exploration of electron-electron correlations. Due to a smaller band gap, one can conduct these experiments using mid-infrared or THz pulses. Noncollinear high-harmonic spectroscopy deepens our understanding of light-induced phenomena in quantum materials and lays the groundwork for engineering next-generation optoelectronic and nanophotonic devices. By harnessing strong light-matter interactions, our work opens new frontiers in exploring non-equilibrium phases of matter and designing quantum materials with bespoke functionalities, advancing the broader fields of ultrafast science, modern photonics, and quantum technology.”.

3. “Finally, do I understand correctly that a dephasing time of 0.5 fs is used in their SBE model simulation? Why is it so much shorter than that used in other SBE simulations?”

Our Response: We thank the referee for highlighting this important point, and we agree that providing the dephasing time in our SBE model simulation is crucial. In the previous manuscript, the mention of 0.5 fs dephasing time T_2 was intended to discuss the population dependence and robustness to

decoherence. To clarify, we used a dephasing time of 3 fs, rather than 0.5 fs. This choice is consistent with the timescales relevant to the specific SiO₂ material and experimental conditions (see references [3-6]).

In the revised supplementary materials, we added the following details about the numerical simulations:
“In the simulation, the parameter for the dephasing time T_2 used is 3 fs.”

Referee #2

“Zhang and coauthors present a comprehensive experimental and numerical investigation of strong-field-induced electron and exciton dynamics in the well-known insulator SiO₂—a material extensively studied within the strong-field community. Over the past decade, the optical response of SiO₂ under intense, ultrashort 800 nm laser fields has been widely explored using various experimental techniques, achieving temporal resolutions down to the sub-femtosecond scale.

In this work, however, the authors introduce a novel approach based on high-harmonic spectroscopy, applied while the sample is driven far from equilibrium by an intense, non-resonant pump field in a non-collinear geometry. This technique, previously demonstrated primarily in gas-phase systems, uniquely combines the capability of high-harmonic generation (HHG) to probe conduction-band electron dynamics with the coherent manipulation of the crystal’s harmonic response via a delay-controlled strong-field pump pulse. The resulting data provide an original and compelling perspective on the intricate interplay between light, electrons, and excitons—all contributing to the optical response of SiO₂ in the UV–EUV spectral range.

The manuscript also includes numerical simulations based on semiconductor Bloch equations, as well as a simplified analytical model, both of which support the experimental findings. These theoretical tools successfully reproduce the observed shift in the center-of-mass of the third-harmonic signal (H3) generated by the 400 nm probe pulse across different pump laser intensities. This allows the authors to disentangle the contributions of various physical effects—including the AC Stark shift and the dynamical Franz–Keldysh effect—to the complex light-driven dynamics observed.”

“I find the proposed methodology highly interesting and complementary to existing experimental techniques. The quality of the data—both experimental and numerical—is excellent and convincingly demonstrates the potential of this approach to investigate a broad range of light-induced phenomena. However, the authors concentrate on the analysis of a single harmonic peak (H3), somehow neglecting all the other signals, while still presenting the technique as ultrahigh-order wave mixings without providing a clear explanation. In my opinion, all the datasets and their analysis could be presented with a higher level of detail: it is this lack of clarity, in my opinion, that raises some concerns on the physical interpretation provided by the authors.

Finally, I am also not totally convinced by the authors regarding the possibility of extending this approach to any other quantum material. Please see the attached comments, divided into major and minor points, to better understand what I mean.

In conclusion, I believe the study is original, significant, and holds potential for publication in Nature Communications. Nevertheless, I would like the authors to address all the points listed below and provide a revised version of the manuscript before I can recommend their work for publication.”

Our Response: We thank the Referee for their positive feedback on our manuscript and appreciate the very detailed review and great constructive suggestions for improvement. We are also grateful for your recognition of the originality and significance of our work. We acknowledge your concerns regarding the focus on the analysis of harmonic spectra, the need for a more comprehensive presentation of the datasets, and the applicability of our approach to other quantum materials. We carefully considered and integrated all questions and suggestions into the revised manuscript. Thank you once again for your thoughtful review.

Specific comments from the Referee

1. "Title and abstract: In my opinion, the terms "Channel-resolved" and "Switching of carrier energies" are not well-established or clearly defined concepts and may not be easily understood by non-expert readers. The title may remain as it is, but a clearer explanation should be included in the abstract. Which channels are referred to? What does "switching" mean in this context? On first reading, I interpreted it as a binary ON/OFF process, while the data presented here rather indicate a manipulation—a continuous fine-tuning—of the excited carriers' energy."

Our Response: We thank the Referee for highlighting the important issue regarding the title and the main focus of our work. We appreciate your concern that terms like "Channel-resolved" and "Switching of carrier energies" may not be well-defined or easily understood.

Thus, in the resubmitted manuscript, we focused on the effect of crossover between the red shifts due to excitonic AC Stark effect and dynamic Franz-Keldysh effect, and the blue shifts due to virtual transitions between excitonic and Bloch states, and the ponderomotive effect. We also replaced the term "channel resolved" in the main text with the more common term "**pathway-resolved**", and the term "switching" with the term "**control**".

Our technique is pathway-resolved because the energy shift given by Eq. (1) is proportional to the real part of the cycle-averaged exciton propagator, which we approximated by the difference of level populations divided by the self-energy of a field-driven exciton. In the revised manuscript, we primarily focused on the exciton center-of-mass energy because it exhibits semiclassical behavior, which remains even at high dephasing rates. This technique can also be applied when the signal exhibits quantum interference fringes, which we observed in the signal of other wave mixings. Further development of these observations requires more rigorous analytical treatment and is left for future work.

To address both comments of Referee #1 and Referee #2, we changed the title to "**Noncollinear Harmonic Spectroscopy Reveals Crossover of Strong-Field Effects**" and introduced the changes to the abstract and conclusions. See our reply to comment 2 of Referee #1.

2. "Introduction, page 3 line 5: Why do the authors refer to the "electronic wavepacket center-of-mass" when discussing the temporal evolution of the H3 signal from the probe pulse? While this is certainly

related to electron dynamics, the phrase may suggest that an electron measurement was performed, rather than a photon-based one. To avoid confusion, I suggest rephrasing.”

Our Response: We thank the referee for pointing out this issue. In the resubmitted version of the manuscript, we rephrased this sentence to connect the temporal evolution of the H3 signal and added the reference to the third-order nonlinear susceptibility equation (42), where the self-energies of the Floquet resonances (43) induced by excitonic and Bloch states are featured in the denominators. These self-energies depend on time delay and introduce the corresponding red and blue shifts of the H3 signal. To further reduce the uncertainty in the resubmitted manuscript, we use the terms “**carrier energy**” which includes the total exciton energy and “**harmonic centroid**”, to explicitly distinguish them.

In the revised manuscript, we revised this part as follows: “**For negative time delays, when the majority of carriers remain in the valence band, the third-harmonic signal of the probe pulse features the redshift due to the pump-field induced renormalization of the electronic subsystem: dynamical Franz-Keldysh effect and excitonic Stark effect.**”

3. “Page 5, line 12: This is the first mathematical formula in the text. The authors should define and explain every physical quantity introduced.”

Our Response: We thank the referee for pointing out this issue. We have changed the definitions paragraph to explain all the missing variables. We also fixed a minor mistake in the notation and highlighted the cycle-averaged quantities with an overbar symbol.

In the revised manuscript, the changed text (including the other comment):

$$\Delta E_i^{(2)}(\Delta t) \approx \sum_{\mathbf{k}, j \neq i} \left| \Omega_{ij, \mathbf{K}_1}^{(0,1)}(\Delta t) \right|^2 \operatorname{Re} \left[\frac{\Delta \bar{n}_{ij, \mathbf{K}_1}(\Delta t)}{\Delta \bar{E}_{ij, \mathbf{K}_1}(\Delta t) + i\gamma} \right]. \quad (1)$$

Here, overline symbol means cycle averaging, indices i and j enumerate the excitonic states and electron-hole pairs in the Bloch bands, \mathbf{K}_1 is the kinetic crystal momentum of the carriers in the pump field, $\Omega_{ij, \mathbf{K}_1}^{(0,1)}(\Delta t) = \mathbf{F}_{0,1}(\Delta t) \cdot \bar{\mathbf{d}}_{ij, \mathbf{K}_1}(\Delta t)$ is the envelope Rabi energy, $\mathbf{F}_{0,1}(\Delta t)$ is the pump pulse envelope multiplied by the unit vector in its polarization direction, $\Delta \bar{E}_{ij, \mathbf{K}_1}(\Delta t) = \bar{E}_{i, \mathbf{K}_1}(\Delta t) - \bar{E}_{j, \mathbf{K}_1}(\Delta t)$ is the difference between the cycle-averaged instantaneous exciton energies, $\gamma = 1/T_2$ is the dephasing rate due to carrier-phonon and carrier-carrier interactions beyond the TDHF approximation, $\Delta \bar{n}_{ij, \mathbf{K}_1}(\Delta t) = \bar{n}_{i, \mathbf{K}_1}(\Delta t) - \bar{n}_{j, \mathbf{K}_1}(\Delta t)$ is the difference between the cycle-averaged populations, which can be found by transforming the conduction (c) and valence (v) band populations in the electronic representation as follows

$$n_{i,\mathbf{K}_1}(\Delta t) = \sum_{\substack{c,v, \\ \mathbf{k}' \rightarrow \mathbf{K}_1}} |A_{c\mathbf{k}'}^{(i)}|^2 n_{c,\mathbf{k}'}(\Delta t) [1 - n_{v,\mathbf{k}'}(\Delta t)],$$

$A_{c\mathbf{k}}^{(i)}$ are the coefficients of exciton state expansion obtained via solution of the Bethe-Salpeter equation [see SI, Eqs. (3) and (4)], $\mathbf{d}_{ij,\mathbf{K}_1}(\Delta t) = \langle i, \mathbf{K}_1 | i \nabla_{\mathbf{K}_1} | j, \mathbf{K}_1 \rangle$ is the transition matrix element. For excitonic states, it can be expressed via the transform of interband matrix elements

$$\mathbf{d}_{ij,\mathbf{K}_1}(\Delta t) = \sum_{\substack{c,v, \\ \mathbf{k}' \rightarrow \mathbf{K}_1}} A_{c\mathbf{k}'}^{(i)*} A_{c\mathbf{k}'}^{(j)} \mathbf{d}_{c\mathbf{k}'}.$$

4. “When the H3 spectrum is discussed for the first time (page 5), the authors state that the trace exhibits an “asymmetrical spectrum profile.” However, the data in Fig. 2 do not clearly show this asymmetry. The spectrum at large negative delays appears rather symmetrical. Is this asymmetry significant? If so, the authors should present data that clearly illustrate this and explain the underlying physical cause.”

Our Response: We thank the Referee for the insightful comment regarding the discussion of the asymmetrical spectrum profile of the H3 spectrum. We agree that Fig. 2 may not clearly illustrate this asymmetry. To address this, we have rephrased this section, and the asymmetrical spectrum profile is now presented more clearly and discussed in detail later in Fig. 4 of the manuscript. In that section, we provide a more thorough analysis and relevant data to support our discussion of asymmetry.

In the revised manuscript, we updated the description to: “The trace exhibits a delay-dependent energy modulation of the spectrum profile, which varies as the delay Δt is tuned from negative to positive values.”.

5. “Page 5, lines 24-28: The authors briefly comment on the differences between the time-resolved harmonic spectra of the pump field and the H3 signal of the probe. I believe this is a key point in the manuscript and deserves more detailed discussion. Is the observed difference due to the fact that, in the case of the H3 signal from the probe, strong-field pumping is involved, while in the other case (switching pump and probe pulses) the actual pump is less intense? Also, intensity values used in Supplementary Fig. 2 are not reported. Alternatively, could the H3 signal be more sensitive to the conduction band minimum and excitonic response—regions where most pump-induced phenomena occur—while the pump harmonics are not? In my view, this point is crucial for understanding the manuscript and should be significantly expanded.”

Our Response: We thank the Referee for pointing out this important issue and for the valuable suggestions. We agree that it is crucial to provide a detailed clarification regarding the intensity regime for the two pulses. The intensities of the two spectra (Fig. 2 and Supplementary Fig. 2) are indeed the same. Since our noncollinear method involves spatially resolved HHG spectra, we can resolve all time-resolved harmonic spectra for both pulses in our measurements.

Generally, the three-photon process has a much higher transition probability than the six-photon process, which is one of the main reasons we chose 400 nm as the probe pulse and 800 nm as the pump field. Also, even-photon transitions in the vicinity of the Γ -point are forbidden by selection rules, which even further reduce the probability of transition due to the pump pulse.

In the revised manuscript, we added a discussion on the exploration of the intensity regime: “We also captured the time-resolved spectra of the pump field, as depicted in supplementary Fig. 2, where we observed an increase in the yields of the harmonics (7th to the 11th order) without any significant centroid energy shift. This behavior contrasts with the energy-modulated spectral profiles observed in harmonics generated from the probe field. Recent work has highlighted the enhanced efficiency of HHG through a similar two-color field in silica⁵². The noted distinction between the harmonics produced by the weaker probe and the intense pump field underscores the field strength-dependent dynamics and responses of the system during the HHG process.”.

We also added the intensity of two pulses in SI when discussing the HHG spectra from the pump pulse: “when intensity of the probe and pump pulse intensities set at 2×10^{11} W/cm² and 4×10^{12} W/cm², respectively.”.

6. *“Page 6, lines 6-9: Are the authors referring to the numerical trace in Fig. 2c (rather than 2b, as written)? If so, the term “acquired” is misleading and should be replaced. Otherwise, the discussion becomes confusing, since Fig. 2b was already introduced on the previous page. Additionally, the authors should explain why the pulse intensity values reported here differ from the initial values provided, especially for the probe pulse.”*

Our Response: We thank the Referee for the careful review and the corrections. We have corrected the reference to Fig. 2c accordingly and replaced the term “acquired” with “simulated” to avoid any confusion between experimental data and simulation results.

Regarding the discrepancy in laser intensities, we apologize for the oversight. For the simulated data in Fig. 2c, we have used comparable pump and probe laser intensity values consistent with those previously reported.

In the revised manuscript, we rephrased the description of laser intensity to clarify that: “Figure 2c depicts the time-resolved data traces of the H3 signal from the probe pulse, simulated with similar probe and pump pulse intensities as those of the experimental data in Fig. 2b using a dephasing time $T_2 = 3$ fs.”

7. *“Page 6, line 9: The authors mention an “observed bleaching effect” at temporal overlap, apparently reproduced by the numerical data. However, when comparing Figs. 2b and 2c, the bleaching is much more pronounced in the numerical trace. Moreover, a secondary contribution is evident, showing an*

opposite temporal evolution (blueshift at negative delays, redshift at positive ones). Can the authors clarify this? A more thorough discussion of the bleaching effect should be included.”

Our Response: We thank the Referee for raising this important point. Indeed, there are some minor differences in the bleaching effect between the data and the simulations. As the delay approaches zero for the two pulses, the strong competition between AC Stark/dynamic Franz-Keldysh effects and ponderomotive effects influences the carrier populations in the ground state and the excitation rate, which ultimately affects the yields of HHG from the probe pulse. Additionally, as shown in Supplementary Fig. 6, the dephasing time also has a significant impact on the bleaching effect.

Regarding the secondary contribution in the numerical trace, as we stated in the manuscript, the main feature of the delay-dependent modulated spectra of H3 arises from the interplay of excitonic AC Stark/dynamic Franz-Keldysh effects and ponderomotive effects. At negative delays, the dominant feature is attributed to the excitonic AC Stark/dynamic Franz-Keldysh effects. Some electrons can be excited to the conduction band, resulting in a ponderomotive blue shift. On the other hand, at positive delays, the redshift occurs under similar conditions. In the experimental setup, the limited signal-to-noise ratio (at the order of a few hundred) affects what we saw, whereas theoretical simulations can be analyzed effectively using a logarithmic scale plot (with signal-to-noise ratio at around ten orders of magnitude).

In the revised manuscript, we recalculated Fig. 2c by adjusting the pump intensity to better align with the experimental spectra. Also, we added a discussion of the differences regarding the higher-order opposite energy shifts as follows: “**While higher-order contributions remain evident, they exhibit an opposite temporal evolution, resulting in a blueshift at negative delays and a redshift at positive delays. However, these effects are not experimentally observable due to the constrained signal-to-noise ratio of the measurements.**”

8. “Page 6, line 20: The perturbation of Bloch and excitonic states is attributed to “charge carriers excited by the pump field.” What kind of excitation is this? Are real carriers excited via photon absorption, or is this a virtual process? What are the initial and final states involved? Given SiO₂’s large band gap, a real excitation would require absorption of at least six photons—an inherently low-probability process. This seems inconsistent with the field being described as non-resonant (see Fig. 3 caption).”

Our Response: We thank the Referee for the comment. The analytical expression of the energy shift (1) is based on the normalized instantaneous population difference of the initial and final states. This term includes contributions of both virtual and real carrier transitions due to interaction with the pump field. The relative contribution of virtual and real processes depends on the dephasing time T_2 . Separation of virtual and real transitions requires consideration of perturbation theory over the pump field, which is complicated due to the high order of excitation processes involved and goes out of the scope of the present

work. On the other hand, the excitation probability of charge carriers by the pump pulse with intensities of $I_l \approx 10^{13}$ W/cm² is significant enough to be detectable by our measurements.

9. “Page 6, line 21: After attributing the effect to charge carriers, the authors introduce a model based on the independent particle approximation, which seems contradictory. Is this approximation justified in this context? The authors should clarify this point.”

Our Response: We developed the adiabatic perturbation theory in the independent particle approximation to simplify the analytic expression by neglecting the Coulomb coupling between the different k points.

In the revised manuscript, we derived the similar expression including excitonic effects, changed the notation, and added a similar expression in the excitonic basis:

$$\Delta E_i^{(2)}(\Delta t) \approx \sum_{\mathbf{k}, j \neq i} \left| \widehat{\Omega}_{ij, \mathbf{K}_1}^{(0,1)}(\Delta t) \right|^2 \operatorname{Re} \left[\frac{\Delta \bar{n}_{ij, \mathbf{K}_1}(\Delta t)}{\Delta \bar{E}_{ij, \mathbf{K}_1}(\Delta t) + i\gamma} \right], \quad (1)$$

Here, the overline symbol means cycle averaging, indices i and j enumerate the excitonic states and electron-hole pairs in the Bloch bands, \mathbf{K}_1 is the kinetic crystal momentum of the carriers in the pump field, $\widehat{\Omega}_{ij, \mathbf{K}_1}^{(0,1)}(\Delta t) = \mathbf{F}_{0,1}(\Delta t) \cdot \bar{\mathbf{d}}_{ij, \mathbf{K}_1}(\Delta t)$ is the envelope Rabi energy, $\mathbf{F}_{0,1}(\Delta t)$ is the pump pulse envelope multiplied by the unit vector in its polarization direction, $\Delta \bar{E}_{ij, \mathbf{K}_1}(\Delta t) = \bar{E}_{i, \mathbf{K}_1}(\Delta t) - \bar{E}_{j, \mathbf{K}_1}(\Delta t)$ is the difference between the cycle-averaged instantaneous exciton energies, $\gamma = 1/T_2$ is the dephasing rate due to carrier-phonon and carrier-carrier interactions beyond the TDHF approximation, $\Delta \bar{n}_{ij, \mathbf{K}_1}(\Delta t) = \bar{n}_{i, \mathbf{K}_1}(\Delta t) - \bar{n}_{j, \mathbf{K}_1}(\Delta t)$ is the difference between the cycle-averaged populations, which can be found by transforming the conduction (c) and valence (v) band populations in the electronic representation as follows

$$n_{i, \mathbf{K}_1}(\Delta t) = \sum_{\substack{c, v, \\ \mathbf{k}' \rightarrow \mathbf{K}_1}} \left| A_{c\mathbf{k}'}^{(i)} \right|^2 n_{c, \mathbf{k}'}(\Delta t) \left[1 - n_{v, \mathbf{k}'}(\Delta t) \right],$$

$A_{c\mathbf{k}}^{(i)}$ are the coefficients of exciton state expansion obtained via solution of the Bethe-Salpeter equation [see SI, Eqs. (3) and (4)], $\mathbf{d}_{ij, \mathbf{K}_1}(\Delta t) = \left\langle i, \mathbf{K}_1 \left| i \nabla_{\mathbf{K}_1} \right| j, \mathbf{K}_1 \right\rangle$ is the transition matrix element. For excitonic states, it can be expressed via the transformation of interband matrix elements

$$\mathbf{d}_{ij, \mathbf{K}_1}(\Delta t) = \sum_{\substack{c, v, \\ \mathbf{k}' \rightarrow \mathbf{K}_1}} A_{c\mathbf{k}'}^{(i)*} A_{c\mathbf{k}'}^{(j)} \mathbf{d}_{c\mathbf{k}'}$$

10. “Page 8, lines 9-16: The discussion of the analytical model lacks clarity. It would be very helpful to plot the temporal evolution of both terms (Eqs. 3 and 4), along with their sum and the pump pulse profile. Also, the statement “when $\Delta t > 0$, the field strength increases” is unclear—isn’t the pump field

temporally symmetric? The authors also state that “a significant number of electrons are excited to conduction bands.” Can they provide an estimate? Lastly, processes such as “exciton ionization” and “interband tunneling of electrons” should be described in greater detail, considering their importance for interpreting the system’s complex optical response under strong fields.”

Our Response: We thank the referee for highlighting the need for clarity in our discussion of the analytical model and for the valuable suggestions. We agree that distinguishing the two contributions is essential for understanding the physical meaning of analytic equation (2), which includes both the blue shift (3) and red shift (4) corresponding to the distinct cases of $\Delta t > 0$ and $\Delta t < 0$. In the revised manuscript, we updated Supplementary Fig. 9 to include the temporal evolution of both terms, along with their sum and the pump pulse profile. On the estimation of the excitation probability: usually in the regime of our experiments, SBE suggested the occupation probability at the order of 10^{-3} . However, it is well known that 1D simulation tends to overestimate the actual excitation probability compared to real 3D simulations. In our work, the “significance” is a relative measure, to be compared in between the cases of negative and positive delays

Regarding the symmetry of the pump field, we apologize for any misunderstanding. The observed asymmetry of the energy shifts is not due to the chirp of the pump pulse, as stated in our response to Referee #3. It comes from the asymmetry of the energy shift peaks $\max_{\Delta t} \Delta E_{1A_1}^{(\pm)}(\Delta t)$ defined via numerical optimization of the main contributions to energy shifts [see SI, Eqs. (30) and (31)]:

$$\Delta E^{(+)}(\Delta t) \approx F_{0,1}^2(\Delta t) \frac{E_{\text{ex}}}{E_{\text{ex}}^2 + \gamma^2} \int_{\text{BZ}} \frac{d^3 k}{(2\pi)^3} \left| \mathbf{d}_{\text{cv} 1A_1, \mathbf{K}_1} \right|^2 \Delta n_{\text{cv} 1A_1, \mathbf{K}_1}(\Delta t),$$

$$\Delta E^{(-)}(\Delta t) \approx F_{0,1}^2(\Delta t) \frac{E_{\text{g}}}{E_{\text{g}}^2 + \gamma^2} \int_{\text{BZ}} \frac{d^3 k}{(2\pi)^3} \left| \mathbf{d}_{1A_1 0, \mathbf{K}_1} \right|^2 \Delta n_{1A_1 0, \mathbf{K}_1}(\Delta t).$$

The contributions from higher states to the red shift are smaller due to the increase of energy in the denominator and smaller dipole matrix elements.

As illustrated in the **Response Letter Fig. 1**, the calculated energy correction amplitudes diverge more significantly at higher pump field amplitudes [see SI, Eqs. (29) and (30)].

Response Letter Fig. 1 | Peaks of energy shifts versus the pump field amplitude.

In the revised manuscript, we added this figure as **Supplementary Fig. 9b** and clarified this section in the main text as follows: “We observed that, under moderate pump amplitudes, the blue and red shifts of the harmonic centroid energy exhibit symmetric growth. However, as the pump intensity increases, the centroid energy shift displays a squeezed asymmetric profile, with the blue shift being larger than the red shift. This anisotropy can be understood as a consequence of asymmetry between the coefficients of perturbative expansion for transitions from the ground state and the first excited state. They are given by the ratios of the squared modulus of the transition matrix element divided by the energy difference of the virtual transition. The higher the pump field intensity, the more significant the asymmetry between the red and blue shifts.”.

11. “Page 11, lines 26-28: This final section is very interesting, but the comparison between a bulk SiO₂ crystal and a 2D material seems questionable. In a 2D material, the substrate indeed strongly affects the optical response, while in a bulk crystal, its influence on the volume-averaged THG signal should be far less significant. The comparison is not well justified.”

Our Response: We thank the Referee for raising this point and apologize for the misunderstanding. In the revised manuscript, we moved this discussion of 2D materials to the Conclusions and Outlook section, as a highlight for possible developments and future perspectives.

We would like to clarify that both experimental techniques and numerical simulations can be effectively applied to 2D materials. The key point is the ratio between the bandgap and the incoming photon energy, not the bandgap energy only. In addition, the quartz and 2D transition metal dichalcogenides (TMDs) both demonstrate excitonic effects, with binding energies around ~100 meV for a 1.5 eV bandgap (quartz has an exciton binding energy of 1.2 eV with an optical bandgap of 9 eV). Second, high-harmonic generation has been successfully achieved in 2D materials using mid-infrared pump pulses [7, 8]. On the other hand, we agree that we cannot currently perform experiments with 800-400 nm two-color pump-probe field using mid-infrared pulses. Although we utilize a 20 μm bulk crystal in experiments, in our

simulations, we neglect propagation effects, which would require numerical integration of the Maxwell or wave equations coupled with semiconductor Bloch equations, as we believe they contribute minimally under our experimental setup. This indicates that our numerical tools are well-suited for studying 2D materials.

In the revised manuscript, we moved this part to the Conclusions and Outlook section: “This experimental approach can be generalized to other systems, such as wide-bandgap two-dimensional (2D) materials, such as nitrides⁵², transition metal dichalcogenides (TMDs)⁵³, which exhibit optical properties similar to those of α -quartz crystal and are characterized by strong excitonic effects. HHG has been successfully demonstrated in 2D materials^{16,17}, making such experiments feasible for further exploration of electron-electron correlations.”.

12. “The authors claim that their framework is general and applicable to quantum materials. I would temper this statement. While the work is certainly valuable and insightful, its applicability to materials beyond wide-gap insulators is not obvious. SiO₂, with its clear excitonic resonance and large band gap, allows observation of specific nonlinear effects that may not appear—or may manifest very differently—in materials with smaller band gaps, such as TMDs. In such systems, the balance between excited carriers and excitons could shift dramatically, altering the underlying physics. This should be acknowledged more cautiously.”

Our Response: We appreciate the Referee's careful consideration regarding the application to 2D materials and acknowledge that these materials have smaller band gaps compared to bulk quartz. As mentioned previously, to observe strong-field nonlinear effects, we can utilize mid-infrared pulses, such as those centered at 0.3 eV. In addition, similar energy modulation for the electronic state has been observed in the WS₂ system by using transient absorption spectra, see Ref. [9].

In the revised manuscript, we moved this discussion to the Conclusion and Outlook section and added one it is possible to apply our experimental and numerical tools to investigate the 2D materials: “Due to a smaller band gap, one can conduct these experiments using mid-infrared or THz pulses.”

13. “Abstract: The phrase “carrier-carrier interactions” is cited as part of “the intricate interplay between light and excited carriers.” However, light is not directly involved in carrier-carrier interactions. I suggest separating light-carrier from carrier-carrier interactions for clarity.”

Our Response: We thank the referee for the suggestion regarding the phrasing in the abstract.

In the revised manuscript, we have rewritten this section to clarify the distinction between light-carrier and carrier-carrier interactions. The revised text in the abstract is: “Yet, the complex interplay between light and excited carriers—via mechanisms such as the AC Stark effect, field-induced coupling of

excitonic and Bloch states, the dynamic Franz-Keldysh effect, and the pondermotive effect—continues to challenge our understanding of quantum systems driven far from equilibrium.”

14. *“Abstract: The material under investigation (SiO₂) is only mentioned at the end of the abstract (line 31). This should be stated earlier (e.g., at line 22) to avoid suggesting that the findings apply universally.”*

Our Response: We thank the referee for the helpful suggestion regarding the placement of the material under investigation (SiO₂) in the abstract. In the revised manuscript, we have mentioned of SiO₂ earlier in the abstract text for clarity. The revised sentence now reads: *“Here, we establish non-collinear high harmonic spectroscopy as a powerful quantum pathway-resolved technique for initiating, tracking, and steering femtosecond carrier dynamics across the energy landscape in the dielectric SiO₂ crystal.”*

15. *“Introduction, line 2: Remove “The” before “light-matter interaction”.”*

Our Response: We thank the referee for pointing out this error. In the revised manuscript, we removed *“The”* before *“light-matter interaction”* as suggested.

16. *“Introduction, lines 4-6: The sentence implies that few-cycle pulses can only be generated via light pulse synthesis, which is inaccurate. Common methods (e.g., hollow-core fibers and chirped mirrors) also apply. Also, given the 10–20 cycle pulses used in this study, is this even relevant? Consider rephrasing.”*

Our Response: We thank the referee for the suggestion. We agree that the original sentence may have implied an exclusive connection between few-cycle pulses and pulse synthesis techniques, which is not closely related to the current topic. Moreover, as our study focuses on multi-cycle pulses (10 cycles), the discussion of few-cycle fields is not essential to the main scope of this work. We have therefore removed the sentence and revised the paragraph to focus more directly on ultrafast light-matter interaction relevant to our study.

In the revised manuscript, we updated version is shown below: *“Strong light-matter interaction creates non-equilibrium states of matter, presenting a remarkable avenue to explore microscopic phenomena and macroscopic properties beyond the reach of conventional materials^{1,2}. Controlling electron dynamics in solids on femtosecond (1 fs = 10⁻¹⁵ s) to attosecond (1 as = 10⁻¹⁸ s) timescales holds great potential for advancing ultrafast optoelectronics³ and exploring non-equilibrium many-body physics with unprecedented accuracy.”*

17. *“References: I suggest removing Ref. 25 unless there’s a citable arXiv version of the manuscript.”*

Our Response: We thank the referee for the suggestion. In the revised manuscript, *we deleted this reference.*

18. *“Page 2, lines 29–33: I recommend citing the work of Sidiropoulos et al., Nat Commun 14, 7407 (2023), which is highly relevant to the discussion of many-body effects induced by strong light fields.”*

Our Response: We thank the referee for the suggestion. We agree that this study by Sidiropoulos et al. (Nat. Commun. 14, 7407 (2023)) provides valuable experimental insights into strong-field-induced many-body dynamics, particularly the role of electron–phonon interactions in driven materials.

In the revised manuscript, we have incorporated this reference and rephrased this paragraph in the main text [44], which now reads as follows: “On the other hand, in the presence of a strong electric field, significant amounts of charge carriers could be excited into the high-energy states, leading to the manifestation of various many-body effects, such as Auger scattering^{35,36}, avalanche breakdown^{37,38}, carrier-carrier^{39–41} and carrier-phonon scattering^{42–44}, etc. Notably, under intense photoexcitation, electron–phonon coupling can dominate the relaxation pathways of carriers, leading to enhanced optical conductivity⁴⁵. The intricate interplay among particle dynamics, excitonic effects, and phonon interactions produces a highly nonlinear and complex evolution of the system. De-spite these advancements, achieving precise control over electronic dynamics and decomposing these constituent effects remains a significant challenge.”.

19. *“A period is missing at the end of the Introduction.”*

Our Response: We thank the referee for pointing out this error. In the revised manuscript, we have added the missing period at the end of the Introduction.

20. *“Figure 1: While the technique enables temporal manipulation of harmonic emission via a pump pulse, the presented datasets are not temporally resolved. The authors should state in the caption that the pump-probe delay is set to 0. Also, correct “upwards” to “downwards” in line 5, if that’s what the sketch suggests. Additional typos: “it’s” and “fields”.”*

Our Response: We appreciate the referee's careful consideration of the misunderstanding and typos.

In the revised manuscript, we have added the rephrased sentence by fixing the typos: “while the beam points **downwards** at an angle of $\theta \sim 18$ mrad, forming a plane parallel to the entrance slit of the XUV spectrometer. **b and c** Ultra-high-order wave-mixing spectra generated from the superposition of an intense fundamental (800 nm) and **the** second harmonic ω_2 weaker **field** (400 nm) in the α -quartz sample.”

21. *“Page 5, line 30: Typo in “noncolinear”.”*

Our Response: We thank the referee for indicating the typo. In the revised manuscript, we corrected “noncolinear” to “noncollinear”.

22. *“Page 7, Equation 1: The term “ $F_0 I(\Delta t)$ ” should be explained just like the others.”*

Our Response: We thank the referee for the suggestion.

In the revised manuscript, we rephrased this part as: “ $\mathbf{F}_{0,1}(\Delta t)$ is the pump pulse envelope multiplied by the unit vector in its polarization direction.”

23. *“Page 8, line 16: This sentence repeats what was written two lines earlier—consider removing or rephrasing.”*

Our Response: We thank the referee for pointing out this misunderstanding description.

In the revised manuscript, we rephrased this sentence as: “When $\Delta t > 0$, a significant number of electrons are excited to the conduction bands, either by exciton dissociation or via interband transitions.”

24. *“Page 9, lines 19–20: The authors note good agreement between experiment, simulation, and analytical models in the power-scaling results. While the amplitudes align, there is a subtle but interesting difference in the temporal behavior. The analytical trace is symmetric about time zero, while in the other two cases, the COM shift reverses for negative delays. This discrepancy might point to underlying physical effects not captured by the analytical model and deserves a brief comment.”*

Our Response: We thank the Referee for the valuable observation regarding the temporal behavior in our power-scaling results between the experimental and analytical solutions. We think this discrepancy arises from the higher-order processes, such as an increase of the transition probability due to the AC Stark and dynamic Franz-Keldysh effects. From the mathematical perspective, the higher-order corrections to the density matrix contain multiple integrations over time, their contribution grows with increase of the pump field intensity, which result in the negative time shifts. These time shifts exist in experimental data and numerical simulations, but they are absent in the simple analytical expression (1). Notably, we employed the time delay steps of 5 fs and 2 fs, while the analytical solution focuses on higher precision in the delay steps and only considers the second-order energy shift at zero delay regarding the crossover of the strong-field effects. We used a simplified analytical approach on purpose to highlight these phenomena.

In the revised manuscript, we included a sentence to discuss this temporal behavior: “Additionally, a subtle temporal behavior is observed in the analytical trace, which is symmetric about zero-time delay. In contrast, the centroid energy shift reverses for negative delays in the other two cases, a discrepancy that can be attributed to higher-order effects.”

25. *“Page 10, line 11: It is unclear what maximum was used for normalizing the curves, as none reach 1. Please clarify.”*

Our Response: We thank the referee for pointing out this unclear description. Indeed, there is a small shift in the y-axis that is not precisely aligned due to some errors in the center of mass calculations.

In the revised manuscript, we updated **Figure 4d** to clarify this normalization process.

26. “Page 11, lines 4–9: The method used to extract the three quantities shown in Fig. 4 d–f is unclear. Consider explaining it in more detail or providing a supporting graph in the Supplementary Information.”

Our Response: We thank the referee for this valuable suggestion.

In the revised manuscript, we added a supporting Fig. 5 (Response letter, Fig. 2) in the Supplementary Information to clarify the normalization process for Figures 4d–f. We hope this additional detail enhances the clarity of our methodology: “To better understand our main figures (such as Fig. 2, Fig. 4, and Fig. 5) presented in the main text, we use the experimental data from Fig. 2 as an example to define the harmonic centroid energy, the bleaching effect, and their modulation depth, as illustrated in main Fig. 4. As shown in Supplementary Fig. 5a, we first select the H3 harmonic from the spatially resolved harmonic spectra (Fig. 1b) and plot it at different time delays. We then integrate the 3D data by summing it along the divergence angle direction to obtain a time-resolved spectrum, as shown in Supplementary Fig. 5b. For this 3D data, we perform the centroid analysis and energy domain analysis, resulting in Supplementary Figs. 5c and 5d, respectively. We can then extract the maximum harmonic yields and energy depth, as indicated by the double arrows.”.

Response Letter Fig. 2 | Procedure for extracting the energy centroid and harmonic yields from time-resolved HHG measurements. **a.** THG spectra of the probe pulse at different time delays Δt . **b.** Integrated time-resolved THG trace for panel **a**, summed over the dispersion angle axis. **c** and **d**, Extracted harmonic yields and harmonic centroid energy variation at different time delays for panel **b**. The double arrows indicate the extracted modulation yields and energy depths presented in Figures 4e and 4f, respectively.

27. *“Page 11, line 26: “Atomically thin” and “two-dimensional” sound redundant—consider removing one.”*

Our Response: We thank the referee for the valuable suggestion.

In the revised manuscript, we revised this section as follows: *“This experimental approach can be generalized to other systems, such as wide-bandgap two-dimensional (2D) materials.”*

28. *“SI, page 11, Fig. 5 caption: Are the authors plotting static absorption, as written in the caption, or optical conductivity, as labeled on the graph? Also, specify whether it is the real or imaginary part.”*

Our Response: We thank the referee for pointing out this issue.

In the revised SI of the manuscript, we have revised this part as: *“b, Calculated real part of the optical conductivity varied at different Coulomb interaction amplitudes.”*

29. *“SI, page 12, line 205: “Normalized to maximize...” what? The sentence is incomplete.”*

Our Response: We thank the referee for pointing out this issue.

In the revised manuscript, we clarified that the caption reads: *“The color bars are normalized to their maximum harmonic yields and presented on a logarithmic scale.”*

30. *“SI, page 15, line 269: Are the assumptions “charge carriers only excited by the pump pulse” and “the probe pulse is short” justified if the probe pulse can generate harmonics and has a pulse duration of 28 fs (~21 optical cycles)? Please comment on this.”*

Our Response: We thank the referee for highlighting this important point.

Regarding the first assumption: the pump pulse is about 20 times stronger in energy compared to the probe. The duration of the pump is slightly short as well (25 fs vs 28 fs). Although the probe pulse has shorter carrier wavelength, first order estimation results in the higher number of carriers excited by the pump pulse compared to that excited by the probe pulse.

Regarding the second assumption: the referee is indeed absolutely right regarding the pulse duration. It is not a Dirac delta function, but if we could simplify it as a Dirac delta function, it would allow us to simplify the analytical solution of the redshift and blueshift, which is very important to reach a conclusion. We agree that an integral over the duration would be better. This assumption could easily be one component contributing to the slight inconsistency you found in your question 24. Overall, we regard this as a higher-order effect.

In the revised SI, we corrected this assumption to: *“For simplicity, we assume that the charge carriers [19] are excited mainly by the pump pulse and neglect the strong-field effects, e.g., intraband acceleration and energy renormalization, induced by the probe pulse.”*

31. *“SI, page 18, line 334: “Reference not found”—please correct.”*

Our Response: We thank the referee for bringing this important part attention. In the revised manuscript, the statement containing this reference was removed.

32. *“Although no usable code is provided, I am not an expert in numerical simulations, but I believe the reference provided by the authors and the supplementary material should contain all the information needed to reproduce the numerical results reported in the manuscript.”*

Our Response: We thank the Referee for this important suggestion. Numerical simulations are crucial for understanding our work, and we believe we have provided detailed references and parameters in the SI that offer the necessary information for reproducing the numerical results reported in the manuscript.

In the revised SI, we specified the dephasing time T_2 and the other parameters: *“In the simulation, the dephasing time T_2 is assumed to be 3 fs, and the grid step sizes for time and real space are 0.0049 mm and 0.048 fs, respectively, covering a range from -10 mm to 10 mm and -100 fs to 100 fs.”*

Referee #3

“The present manuscript by Zhang et al. reports on an experimental study of non collinear high harmonic generation, the purpose of which was to probe temporal carrier dynamics in an alpha-quartz crystal. The authors investigated the effects on the third harmonic (H3) energy of the probe beam (400 nm) as a function of the delay between pump (800 nm) and probe beams. Their observations included a red shift of the H3 signal for negative time delays, followed by a blue shift at positive time delays. The authors reproduce this experimental observation through two means: firstly, through numerical simulations (SBE); secondly, through an analytical derivation (adiabatic perturbation theory). Furthermore, the authors observe a bleaching effect on the H3 signal when the two beams are temporally overlapped (time delay equal to zero). The study goes on to consider the dependencies of these two effects as a function of the pump power intensity, claiming the possibility to manipulate the electron energies by the driving field intensity. The report also presents a simulated dependence of the CoM depth on the electron-hole Coulomb interaction potential, which the authors claim to be useful in the study of many-body effects in solid.”

“I think the work has some interesting parts, like the maths behind eq. 2-3-4 in the main text or the reported experimental data, but there are some things that need to be fixed before it can be published.”

Our Response: We thank the Referee for the comprehensive summary of our manuscript and for the positive feedback on our work. We greatly appreciate the detailed review and constructive suggestions for improvement. We ensured that all questions and suggestions were carefully considered and integrated into the revised manuscript. Thank you once again for your thoughtful review.

“My main concern is about the chirp of the two beams. How do the authors make sure that the beam's chirp isn't affecting what they're seeing? Do they just look at the time-resolved HHG for the pump pulse? My second concern is about a paper recently published in Nat Com that reports very similar studies (<https://doi.org/10.1038/s41467-024-52774-9>). Could the author explain what they are doing differently compared to that paper?”

Our Response: Regarding the chirp of the two beams, as demonstrated in the FROG trace shown in Supplementary Fig. 1, we consistently measure the FROG before commencing our experiments. We compensate for the chirp of both pulses using a glass and grating pair within the coherent laser system, ensuring a chirp-free condition throughout our experiments. Additionally, we closely monitor the wave-mixing signal to verify this compensation, allowing us to focus on the interaction effects without any chirp-induced artifacts.

As for your second concern regarding the recently published paper [2], while both studies investigate high-order harmonic generation, our work distinguishes itself through the specific implementation of two-color non-collinear wave mixing in silica and a detailed analysis of the resulting dynamics. We clarified these distinctions in the revised manuscript to emphasize the unique contributions of our study.

In the revised manuscript, we added a discussion for the recommended *Nature Communications* paper [2]: “We also captured the time-resolved spectra of the pump field, as depicted in supplementary Fig. 2, where we observed an increase in the yields of the harmonics (7th to the 11th order) without any significant central energy shift. This behavior contrasts with the energy-modulated spectral profiles observed in harmonics generated from the probe field. **Recent work has highlighted the enhanced efficiency of HHG through a similar two-color field in silica**⁵¹.”

Specific comments from the Referee

1. “On page 3, line 19, where is the dashed line?”

Our Response: We thank the referee for bringing this to our attention. Indeed, the dashed line was not presented.

In the revised manuscript, we corrected this issue: “Notably, owing to the slight inclination of the two pulses, as shown by the **solid** line in Fig. 1a.”

2. “In figure 1, it would be useful to report the same y-scale for the simulated and measured contour plots. What information can be extracted from the intensity dependence of the HH signals? Lastly, there is a typo in panel c at 14 eV: there should be (1,4) instead of (2,4).”

Our Response: We thank the referee for pointing out these important issues regarding the y-scale and the assignment of the wave-mixing signals. We have updated Figure 1 accordingly.

Regarding the information extracted from the intensity dependence of the HH signals, we have summarized this in Figure 4, indicating that we can control the state up to 100 meV by varying the intensity. Additionally, as shown in Supplementary Figure 2, the scaling confirms the photon dressing effect, which approaches nonperturbative conditions and supports our analysis.

In the revised manuscript, we updated **Figure 1** and corrected the typo in panels b and c to reflect **(1,4) mixing signal instead of (2,4)**.

3. “L. 18 p. 5, it is not specified the crystal direction where the H3 signal of figure 2 was acquired. When comparing to the result in the SI, like fig. 2, it should be important that the direction is the same (Γ -K). Also, have you tried the same experiment but in the other direction (from Γ to M)?.”

Our Response: We appreciate the referee for highlighting these important points regarding the pulse polarization orientation. Yes, we also conducted the same experiment in the Γ to M direction and included the comparison results in the Supplementary Information.

In the revised manuscript, we added a sentence to the caption of Figure 2: “The laser pulses are polarized along the Γ -K orientation of the crystal for both the measured and simulated spectra.”

Response Letter Fig. 3 Crystal orientation dependent tr-THG trace. **a**, Calculated band structure of α -quartz, highlighting one valence band and one conduction band for the Γ -M and Γ -K orientations. **b**, Experimentally extracted delay-dependent energy variation in the Γ -M (red) and Γ -K (blue) orientations.

Additionally, as shown in the Response Letter Fig. 3 (supplementary Fig. 5), we added a figure in Supplementary Information Section 1.5, titled "Dependence of Third Harmonic Spectrum on Crystal Orientations," as follows: “To investigate the impact of nonadiabatic energy shifts of electrons across these orientations, we first calculated the band structure of the quartz crystal using the density functional theory (DFT) and GW many-body perturbation theory (details provided in Section 2.1). Subsequently, we conducted experiments with the electric field strengths of the pump and probe pulses maintained at 3.6×10^{12} W/cm² and 2×10^{11} W/cm², respectively. We recorded the time-resolved harmonic spectra while varying the crystal orientation in the Γ -M and Γ -K direction. The primary results of the time-dependent center of mass energy shift are presented in Supplementary Fig. 4**b**. At negative time delays, the energy shifts exhibit similar magnitudes, indicating that the redshift is comparable for both directions. At positive time delays, the blueshift for the Γ -K orientation are greater than that for the Γ -M orientation. This observation is consistent with lower effective mass and higher optical matrix elements along the Γ -K orientation compared to the Γ -M.”

4. “Fig. 2, panel a: the y-scale doesn't have labels, and the numerical simulation was calculated using pulse intensities that are different from the experimental ones (9 vs. 0.2 PW/cm² for the probe). What does the simulation show with the experimental values?”

Our Response: We appreciate the referee's suggestion to include a label on the y-axis of Fig. 2 to enhance the clarity of the figure.

Regarding the laser intensities, the simulated intensity closely matches the experimental settings. We apologize for the incorrect numbers provided, which resulted from multiple revisions of the submitted manuscript.

In the revised manuscript, we have added the y-axis label in Fig. 2 and also the excitonic level in the band structure of quartz and added one sentence: “The optical properties of α -quartz crystal are significantly influenced by the Coulomb interaction of electrons and holes⁴⁷. This results in the existence of a fundamental exciton with a high binding energy of ^{23,47}, which is self-trapped in the anion complex (SiO₄)⁴⁻ (see inset in Supplementary Fig. 6). According to the ab initio study⁴⁸, the exciton localization length of up to 4.2 Å, which is comparable with the lattice constants, $a = b = 4.9$ Å and $c = 5.4$ Å⁴⁹, implying that the fundamental exciton in α -quartz is of the Frenkel type⁵⁰ and gives the exciton's first states as $1A_1$ ⁵¹.”.

Additionally, we rephrased the sentence as follows: “Figure 2c depicts the time-resolved data traces of the H3 signal from the probe pulse, simulated with similar probe and pump pulse intensities as those of the experimental data in Fig. 2b using a dephasing time $T_2 = 3$ fs.”

5. “In equation 1, please explain what the subscript 1 of K means.”

Our Response: We thank the referee for the suggestion.

In the revised manuscript, we have changed text (including the other comment):

$$\Delta E_i^{(2)}(\Delta t) \approx \sum_{\mathbf{k}, j \neq i} \left| \Omega_{ij, \mathbf{K}_1}^{(0,1)}(\Delta t) \right|^2 \operatorname{Re} \left[\frac{\Delta \bar{n}_{ij, \mathbf{K}_1}(\Delta t)}{\Delta \bar{E}_{ij, \mathbf{K}_1}(\Delta t) + i\gamma} \right]. \quad (1)$$

Here, overline symbol means cycle averaging, indices i and j enumerate the excitonic states and electron-hole pairs in the Bloch bands, \mathbf{K}_1 is the kinetic crystal momentum of the carriers in the pump field, $\Omega_{ij, \mathbf{K}_1}^{(0,1)}(\Delta t) = \mathbf{F}_{0,1}(\Delta t) \cdot \bar{\mathbf{d}}_{ij, \mathbf{K}_1}(\Delta t)$ is the envelope Rabi energy, $\mathbf{F}_{0,1}(\Delta t)$ is the pump pulse envelope multiplied by the unit vector in its polarization direction, $\Delta \bar{E}_{ij, \mathbf{K}_1}(\Delta t) = \bar{E}_{i, \mathbf{K}_1}(\Delta t) - \bar{E}_{j, \mathbf{K}_1}(\Delta t)$ is the difference between the cycle-averaged instantaneous exciton energies, $\gamma = 1/T_2$ is the dephasing rate due to carrier-phonon and carrier-carrier interactions beyond the TDHF approximation,

$\Delta \bar{n}_{ij, \mathbf{K}_1}(\Delta t) = \bar{n}_{i, \mathbf{K}_1}(\Delta t) - \bar{n}_{j, \mathbf{K}_1}(\Delta t)$ is the difference between the cycle-averaged populations, which can be found by transforming the conduction (c) and valence (v) band populations in the electronic representation as follows

$$n_{i, \mathbf{K}_1}(\Delta t) = \sum_{\substack{c, v, \\ \mathbf{k}' \rightarrow \mathbf{K}_1}} \left| A_{cv, \mathbf{k}'}^{(i)} \right|^2 n_{c, \mathbf{k}'}(\Delta t) \left[1 - n_{v, \mathbf{k}'}(\Delta t) \right],$$

$A_{cv, \mathbf{k}}^{(i)}$ are the coefficients of exciton state expansion obtained via solution of the Bethe-Salpeter equation [see SI, Eqs. (3) and (4)], $\mathbf{d}_{ij, \mathbf{K}_1}(\Delta t) = \langle i, \mathbf{K}_1 | i \nabla_{\mathbf{K}_1} | j, \mathbf{K}_1 \rangle$ is the transition matrix element. For excitonic states, it can be expressed via transform of interband matrix elements

$$\mathbf{d}_{ij, \mathbf{K}_1}(\Delta t) = \sum_{\substack{c, v, \\ \mathbf{k}' \rightarrow \mathbf{K}_1}} A_{cv, \mathbf{k}'}^{(i)*} A_{cv, \mathbf{k}'}^{(j)} \mathbf{d}_{cv, \mathbf{k}'}$$

6. “In figure 3, panel b, please explain the meaning of the colour difference between the arrows.”

Our Response: We thank the referee for this comment. The arrows show possible excitation and virtual transition pathways with red and blue shifts. On the left there are excitonic transitions (two-particle processes), on the right there are direct interband transitions (single-particle processes). Colors of arrows are colors of energy shifts.

In the revised manuscript, we added one sentence in the caption of Fig, 3 as: “The colors (red and blue) of the arrows correspond to the energy shifts.”

7. “In figure 4, the error bars in panels e and f do not go along the x-axis, so they cannot be related to fluence uncertainties.”

Our Response: We thank the referee for this important observation. We agree that the error bars in Figure 4 panels e and f do not pertain to fluence uncertainties, and we have removed that reference as it is not applicable in this context.

In the revised manuscript, we removed the previous description and updated it to state: “The error bars represent statistical uncertainties based on five measurements under identical experimental conditions.”

8. “The inset in Fig. 5 is not clear to me, and it should be stated in the caption that this result is due to a simulation and not to an experiment.”

Our Response: We thank the referee for the comment.

In the revised manuscript, we moved the Fig. 5 to Supplementary Fig. 8 and updated the caption to explicitly state that this result is derived from a simulation rather than an experiment: “Simulated carrier energy shift in the presence of different Coulomb potential amplitude.”

9. “Lastly, I wonder if the dephasing time T_2 could be extracted by the experimental traces.”

Our Response: We thank the referee for the insightful question regarding the dephasing time T_2 . We acknowledge that a constant dephasing time can be extracted when comparing the experimental and simulated traces. However, we also note that the dephasing time can depend on factors such as laser intensity and other effects. To ensure accuracy and avoid overstatement, we prefer to be cautious in our claims regarding this extraction.

In the revised manuscript, we added one sentence: “This correlation may also offer a method for directly measuring the dephasing time T_2 in such a complex system.”

10. “In the references related to Science there is always the year 1979, I believe it is a typo.”

Our Response: Thank you for pointing this out. In the revised manuscript, we corrected the publication years of the Science references accordingly.

References

- [1] J. B. Bertrand, H. J. Wörner, H. C. Bandulet, É. Bisson, M. Spanner, J. C. Kieffer, D. M. Villeneuve, and P. B. Corkum, Ultrahigh-order wave mixing in noncollinear high harmonic generation, *Phys. Rev. Lett.* **106**, 023001, (2011).
- [2] S. D. C. Roscam Abbing, N. Kuzkova, R. van der Linden, F. Campi, B. de Keijzer, C. Morice, Z. Y. Zhang, M. L. S. van der Geest, and P. M. Kraus, Enhancing the efficiency of high-order harmonics with two-color non-collinear wave mixing in silica, *Nat. Commun.* **15**, 8335 (2024).
- [3] L.-J. Lü and X.-B. Bian, Ultrafast intraband electron dynamics of preexcited SiO₂, *Opt Express* **28**, 13432 (2020).
- [4] M. Garg, H. Y. Kim, and E. Goulielmakis, Ultimate waveform reproducibility of extreme-ultraviolet pulses by high-harmonic generation in quartz, *Nat. Photonics*, **12**, 291–296 (2018).
- [5] T. T. Luu, M. Garg, S. Yu. Kruchinin, A. Moulet, M. T. Hassan, and E. Goulielmakis, Extreme ultraviolet high-harmonic spectroscopy of solids, *Nature* **521**, 498 (2015).
- [6] M. Garg, M. Zhan, T. T. Luu, H. Lakhota, T. Klostermann, A. Guggenmos, and E. Goulielmakis, Multi-petahertz electronic metrology, *Nature* **538**, 359 (2016).
- [7] H. Liu, Y. Li, Y. S. You, S. Ghimire, T. F. Heinz, and D. A. Reis, High-harmonic generation from an atomically thin semiconductor, *Nat. Phys.* **13**, 262 (2017).
- [8] N. Yoshikawa, T. Tamaya, and K. Tanaka, High-harmonic generation in graphene enhanced by elliptically polarized light excitation, *Science* **356**, 736 (2017).
- [9] Kobayashi, Y., Heide, C., Johnson, A.C. et al. Floquet engineering of strongly driven excitons in monolayer tungsten disulfide. *Nat. Phys.* **19**, 171–176 (2023).

Response to the Referees

Referee #1

“The reviewer has addressed my comments. In particular, I like the new title, abstract, and conclusion. I am also excited to see that the authors have further improved their manuscript by taking the other reviewers' comments into account.”

Our Response: We sincerely thank the Referee for the thorough evaluation of our manuscript, and we are grateful for your recommendation for publication in Nature Communications. Thank you again for your thoughtful review.

Referee #2

“I want to thank the authors for their significant efforts in addressing all the comments and questions raised by the three reviewers.

I firmly believe that the manuscript has improved in clarity and readability. Experimental and simulated datasets are now better presented, and their interpretation is more solid and convincing. Also the Conclusion and Outlook section results to be more precise about future applications of this technique to the study of more complex systems.

In conclusion, I am happy to recommend the actual version of the work for publication in Nature Communications.”

Our Response: We appreciate your recognition of our efforts to address all comments and questions from the reviewers. We are delighted to hear that the manuscript's clarity, presentation of data, and interpretation have improved, as well as the precision of the Conclusion and Outlook section regarding future applications. We are grateful for your recommendation for publication in Nature Communications.

Referee #3

“In the revised version of the paper, Zhang et al. provided detailed and reasonable responses to all the queries raised by the other reviewers.

Regarding my personal comments, I am satisfied with the author's responses. Therefore, I believe that the revised version of the paper meets publication criteria. However, I would suggest improving panel b of Fig. 3 by adding in the caption what the authors mean by 'Up', 'DeltaExs' and 'DeltaEFK' to improve readability.”

Our Response: We appreciate your acknowledgment of our detailed responses to all queries. We are pleased to hear that the referee finds the revised version meets the publication criteria. Thank you again for your thoughtful review.

In response to their suggestion, we improved panel b of Fig. 3 by adding clarifications in the caption for 'Up', 'Delta Exs', and 'Delta EFK' to enhance readability, and read as: “Charge carrier transitions in the presence of a strong, non-resonant electric field result in energy level and band splitting due to the excitonic Stark effect (ΔE_{XS}), and the dynamical Franz-Keldysh effect (ΔE_{FK}) and ponderomotive effect (U_p) within the continuum of Bloch states.”